# The multifaceted role of the inferior colliculus in sensory prediction, reward processing, and decision-making

Xinyu Du[1,2†], Haoxuan Xu[3,4†], Peirun Song[1,2], Yuying Zhai[1,2], Hangting Ye[1,2], Xuehui Bao[3,4], Qianyue Huang[3,4], Hisashi Tanigawa[3,4], Zhiyi Tu[5], Pei Chen[5], Xuan Zhao[5], Josef P Rauschecker[6], Xiongjie Yu[1,2,3,4,5]*

[1]Department of Anesthesia, Women's Hospital, Zhejiang University School of Medicine, Hangzhou, China; [2]Zhejiang Provincial Key Laboratory of Precision Diagnosis and Therapy for Major Gynecological Diseases, Women's Hospital, Zhejiang University School of Medicine, Hangzhou, China; [3]College of Biomedical Engineering and Instrument Science, Zhejiang University, Hangzhou, China; [4]Key Laboratory for Biomedical Engineering of Ministry of Education, Hangzhou, China; [5]Department of Anesthesiology, Shanghai Tenth People's Hospital, Tongji University School of Medicine, Shanghai, China; [6]Department of Neuroscience, Georgetown University, Washington, DC, United States

**\*For correspondence:**
yuxiongj@gmail.com

[†]These authors contributed equally to this work

## eLife Assessment

This **important** study presents a finding on the role of the Inferior Colliculus in sensory prediction, cognitive decision-making, and reward prediction. The evidence supporting the claims of the authors is **convincing**. The work will be of broad interest to sensory neuroscientists.

**Abstract** The inferior colliculus (IC) has traditionally been regarded as an important relay in the auditory pathway, primarily involved in relaying auditory information from the brainstem to the thalamus. However, this study uncovers the multifaceted role of the IC in bridging auditory processing, sensory prediction, and reward prediction. Through extracellular recordings in monkeys engaged in a sound duration-based deviation detection task, we observed a 'climbing effect' in neuronal firing rates, indicative of an enhanced response over sound sequences linked to sensory prediction rather than reward anticipation. Moreover, our findings demonstrate reward prediction errors within the IC, highlighting its complex integration in auditory and reward processing. Further analysis revealed a direct correlation between IC neuronal activity and behavioral choices, suggesting its involvement in decision-making processes. This research highlights a more complex role for the IC than traditionally understood, showcasing its integral role in cognitive and sensory processing and emphasizing its importance in integrated brain functions.

## Introduction

The brain's architecture is a complex hierarchy, where functions become increasingly sophisticated from lower to higher levels. The cerebral cortex, for instance, is pivotal for higher-order cognitive functions such as decision-making, motivation, attention, learning, memory, problem-solving, and conceptual thinking (*Kremkow and Alonso, 2018*; *Miterko et al., 2018*). Conversely, subcortical sensory systems are traditionally viewed as mere conduits for sensory information (*Sherman, 2016*),

despite suggestions of thalamic structures playing roles in advanced perceptual tasks (*Jones, 2012*; *Sherman, 2005*). Yet, the involvement of sensory neurons below the thalamo-cortical level in cognitive behaviors remains an underexplored domain. Within the auditory hierarchy, spanning seven stages from the cochlea to the auditory cortex (AC) (*Pickles, 2015*), the inferior colliculus (IC) emerges as a crucial intermediary, instrumental in processing spatial representations (*Cohen and Knudsen, 1999*; *Ono and Ito, 2015*) and novelty detection (*Malmierca et al., 2009*; *Pérez-González et al., 2005*). Utilizing the oddball paradigm, researchers have identified stimulus-specific adaptation (SSA) within the IC (*Ayala et al., 2016*; *Duque and Malmierca, 2015*; *Malmierca et al., 2009*; *Valdés-Baizabal et al., 2021*; *Valdés-Baizabal et al., 2017*), where neurons decrease their response to frequently occurring sounds while maintaining robust reactions to deviant ones. This phenomenon, indicative of predictive coding, underscores the IC's relevance with sensory prediction. Exploration along the auditory pathway suggests SSA's origination from the IC (*Carbajal and Malmierca, 2018*; *Khouri and Nelken, 2015*; *Song et al., 2023*). Intriguingly, the dynamic interplay between sensory prediction and sensory encoding, particularly within lower sensory systems charged with the faithful representation of external stimuli, remains an underexplored yet captivating area of inquiry.

Traditionally viewed as a key player in auditory processing, the IC is situated in the auditory midbrain and serves as a vital conduit, channeling inputs from the brainstem's auditory nuclei towards the thalamocortical auditory pathway. Intriguingly, the IC is endowed with dopamine receptors and exhibits nerve terminals positive for tyrosine hydroxylase, signifying a channel for dopaminergic inputs originating from the subparafascicular thalamic nucleus (*Harris et al., 2021*; *Nevue et al., 2015*; *Nevue et al., 2016*). This positions the IC as a crucial intersection where auditory processing and reward mechanisms converge. Recent research underscores the significant influence of dopamine on modulating auditory responses within the IC (*Gittelman et al., 2013*; *Hoyt et al., 2019*). This modulation suggests a complex interplay wherein dopamine has the potential to either suppress or enhance neural activity in the IC, depending on various factors (*Gittelman et al., 2013*; *Hoyt et al., 2019*). However, these studies were conducted in rodents, and the existence and role of dopaminergic inputs and reward processing in the primate IC remain underexplored. This amalgamation of sensory and reward coding within the IC paves new paths for research, urging a deeper inquiry into how this midbrain entity integrates auditory and reward information to mold behavioral processes, thereby redefining the traditional auditory-centric view of the IC and opening avenues for understanding its multifaceted role in cognitive and sensory modulation.

In this research, we embarked on a deviation detection task centered around sound duration with trained monkeys, performing extracellular recordings in the IC. Our observations unveiled a 'climbing effect'—a progressive increase in firing rate after sound onset, not attributable to reward but seemingly linked to sensory experience such as sensory prediction. Moreover, we identified signals of reward prediction error and decision-making. These findings propose that the IC's role in auditory processing extends into the realm of complex perceptual and cognitive tasks, reshaping previous assumptions about its functionality.

## Results
### Deviant response dynamics in duration deviation detection

Neurons within IC (*Figure 1—figure supplement 1*) are distinguished by their ability to sustain firing, illustrating a consistent encoding of sound duration through tonic firing throughout the entire auditory length in one example neuron (*Figure 1A*). An analysis of 104 neurons (53 from Monkey B and 51 from Monkey J) demonstrated that these neurons maintained sustained firing responses to sounds of varying durations—100, 500, and 1000 ms (*Figure 1B*). This sustained firing behavior suggests the IC's role in processing temporal aspects of auditory perception, highlighting its potential involvement in functions related to duration perception.

Despite traditionally being viewed as a lower-level nucleus within the auditory pathway in comparison to the AC, and rarely explored from a perceptual standpoint, this study breaks new ground by examining the involvement of IC neurons in duration perception. Monkeys were trained to perform a deviation detection task focusing on duration. *Figure 1C and D* outline the behavioral task, presenting repetitive sounds in blocks with oddball stimuli. Each block consisted of 3–6 repeated white-noise bursts of 300 ms duration at 600 ms interstimulus intervals, concluding with either a standard sound

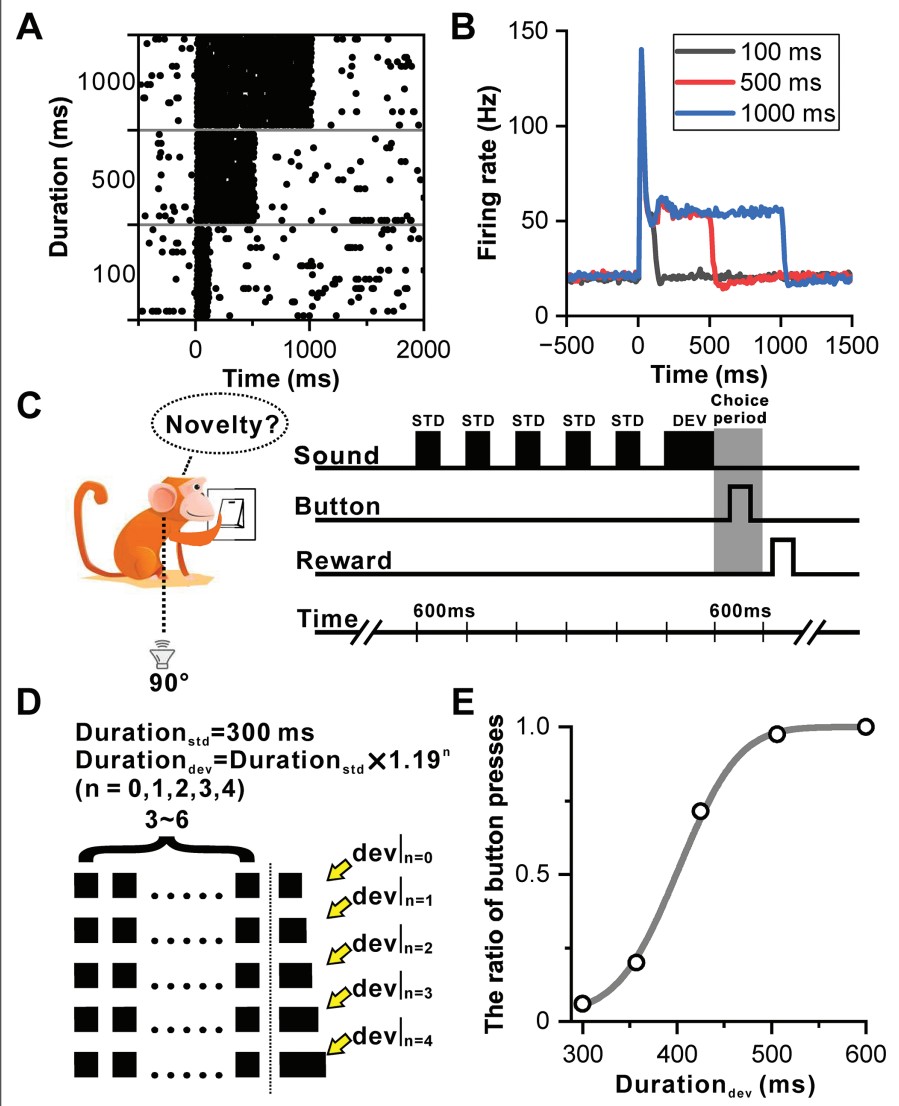

**Figure 1.** Sustained firing in IC and deviation detection paradigm based on sound duration. (**A**) Raster plots of a representative neuron depicting the response profile to white noise stimuli with varying durations (100 ms, 500 ms, and 1000 ms). (**B**) PSTHs of neuronal population responses (n=104) to the stimuli described in (**A**) (black, 100 ms white noise; red, 500 ms white noise; blue, 1000 ms white noise). (**C**) Schematic display of deviation detection behavior. A button is positioned in front of the monkey, and a loudspeaker is placed contralateral to the recording site at the height of the monkey's ear. Within each experimental block, a series of repeated 300 ms white noise bursts (WNB) serves as the standard stimulus, accompanied by a rare-duration WNB serving as the deviant stimulus. Following the presentation of the deviant stimulus, the primate is required to press the designated button within a 600 ms timeframe to obtain a water reward. (**D**) Oddball stimulation paradigm employed in all experimental blocks. While the duration of the standard sound remains constant at 300 ms, the duration of the deviant sound deviates from the standard sound across five distinct levels. This deviation is determined by the formula: $\mathrm{Duration_{deviant}} = \mathrm{Duration_{standard}} \times \lambda^n$ (where n=0, 1, 2, 3, 4; $\lambda = 1.19$). Furthermore, the number of standard stimuli within each block is randomly selected from a range of 3–6. Upon completion of the deviant sound, the primate is required to press the designated button within a 600 ms window to receive the reward. In the absence of a deviant sound (control condition), the reward is granted if the primate refrains from pressing the button within 600 ms after the onset of the final sound. (**E**) Cumulative Gaussian fits of psychophysical data in one example session. The duration of the deviant sound is plotted along the abscissa, while the ratio of button presses is plotted along the ordinate.

The online version of this article includes the following figure supplement(s) for figure 1:

**Figure supplement 1.** Neurons in the inferior colliculus with sustained response.

(control condition) or a deviant sound (*Figure 1D*). Monkeys were required to press a button within 600 ms after the offset of the deviant stimulus to receive a water reward. Correct rejections occurred when the monkey refrained from pressing the button within 600 ms following a standard sound in control trials, where the deviant duration equaled the standard duration. The task difficulty was modulated by setting deviant durations at five levels: 300 (control), 357, 425, 506, and 600 ms (*Figure 1D*), with the pressing ratio approaching zero in the control condition and increasing with the deviant duration, reaching 1 at the 600 ms deviant duration in one example session (*Figure 1E*).

The initial analysis focused on the deviant response during correct trials, revealing that an example IC neuron exhibited continuous firing throughout the sound presentation (*Figure 2A*). Following the transient peak response in the initial 0–60 ms window, the neuron's firing rate began to increase approximately 100 ms after the sound's onset, continuing until the sound's conclusion. This phenomenon, illustrated by aligning the responses to the sound's offset (*Figure 2B*), indicated a proportional increase in firing rate with longer sound durations, suggesting a duration-dependent 'climbing effect'. To quantitatively assess this dynamic, the neuronal firing rate was examined across different sound durations within two distinct temporal windows: the 'onset window' (0–60 ms post-sound onset) and the 'late window' (−100–0 ms relative to sound offset). Notably, the firing rate of the example neuron in the late window escalated with increasing duration (p=6.51e-23, ANOVA with post-hoc test; red in *Figure 2C*), while remaining constant in the onset window (p=0.69, ANOVA with post-hoc test; black in *Figure 2C*), contrasting sharply with the non-behavior condition (*Figure 1B*) which showed no climbing effect.

To assess if the 'climbing effect' was contingent on behavioral context, we compared responses across behavior and non-behavior conditions. Sound durations were analyzed at three levels (100, 500, and 1000 ms for the non-behavior condition as shown in *Figure 1A and B*) and five levels (300, 357, 425, 506, and 600 ms for the behavior condition, illustrated in *Figure 1C and D*). In the same neuronal population (n=99), the firing rate increased after the transient peak response for the behavior situation (red line of *Figure 2—figure supplement 1A*), while the firing rate stayed constant for the non-behavior condition from 100 ms to the end of the sound (blue line of *Figure 2—figure supplement 1A*). The 'Response Dynamic Index' (RDI) was introduced to quantify the climbing effect, calculated as the normalized difference between responses in the late window (−100–0 ms relative to the sound's offset) and the after-peak window (100–200 ms relative to the sound's onset). This comparative analysis indicated a significantly higher RDI in the behavior condition (p=4.40e-19, paired t-test, *Figure 2—figure supplement 1B*), indicating the climbing effect's reliance on behavioral context. Additionally, RDIs across various deviant sounds showed strong correlations (p<0.001, Pearson Correlation Analysis, *Figure 2—figure supplement 2*) and similar values across differing deviants (*Figure 2—figure supplement 2*). Detailed insights into the tonotopic distribution of the climbing effect within the IC are provided in *Figure 2—figure supplement 3*.

## Standard response dynamics in duration deviation detection

Building upon the findings from the deviant responses, we next explored whether the climbing effect also manifested in responses to preceding standard stimuli, thereby examining the influence of sensory prediction and repetition on IC neuronal activity. The same example neuron exhibited the climbing effect post-peak response to standard stimuli, which intensified as successive standard sounds progressed from the second to the seventh stimulation (*Figure 2D*). The firing rate markedly decreased from the first to the second sound in both the late window (p=3.33e-11, paired t-test, red in *Figure 2E*) and onset window (p=2.02e-40, paired t-test, black in *Figure 2E*). The rate then increased in the late window (p=3.05e-5, ANOVA with post-hoc test: second versus fourth, p=0.0398; fourth versus seventh, p=1.00; red in *Figure 2E*) but remained consistent in the onset window (p=0.963, ANOVA with post-hoc test, black in *Figure 2E*) from the second to the seventh sound. Interestingly, no climbing effect was noted in the late window for the first stimulus, and the effect amplified with each subsequent standard presentation.

To monitor the dynamic of the climbing effect across repetitive sound presentations, we analyzed responses to the 300 ms sound across the entire oddball block in population. We normalized the firing rate relative to the response to the first stimulus in the block, considering both the onset and late windows (*Figure 2F*), to account for variability across the neuronal population. In the onset window, responses remained relatively stable from the second to the seventh sound (black line, *Figure 2F*).

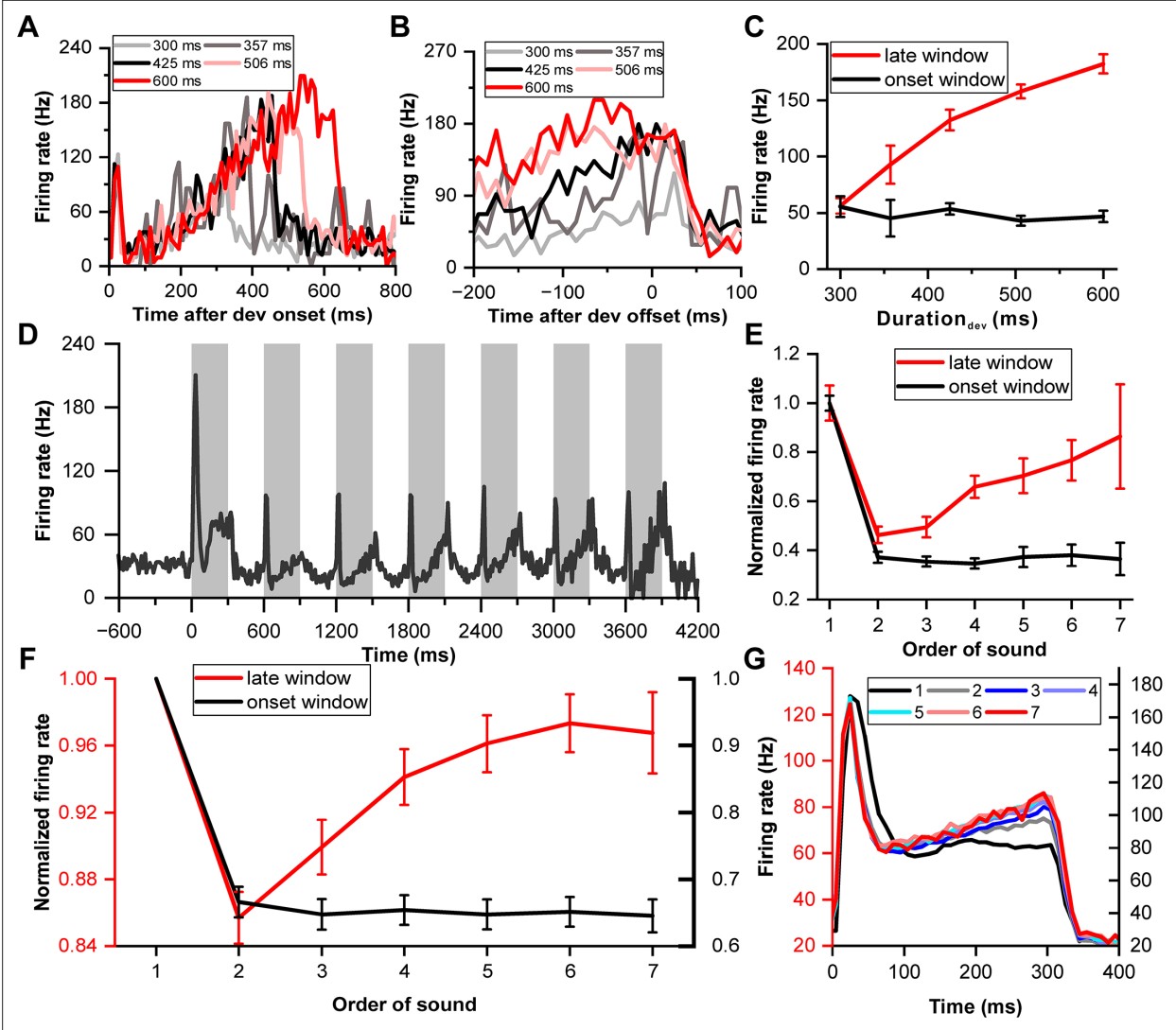

**Figure 2.** Neuronal climbing effect in deviation detection behavior. (**A**) Peri-stimulus time histograms (PSTHs) depicting example neuronal responses to different deviant sounds aligned to deviant onset (light gray, 300 ms deviant sound; gray, 357 ms deviant sound; black, 425 ms deviant sound; pink, 506 ms deviant sound; red, 600 ms deviant sound). (**B**) The same neuronal responses as in (**A**), but this time aligned to the offset of the deviant stimulus. The PSTHs represent the neural activity before the offset of each deviant sound. The color scheme remains consistent with (**A**) to indicate the different deviant durations. (**C**) Firing rate to deviant stimuli in two windows: the late window ([−100 0] ms relative to offset time) and the onset window ([0 60] ms relative to onset time) (red, late window; black, onset window). (**D**) PSTH showing neuronal responses to standard sounds (1-7) from the neuron described in (**A**). (**E**) Normalized firing rate of the neuron described in (**A**) to standard sounds in two temporal windows (late window: [−100 0] ms relative to offset time; onset window: [0 60] ms relative to onset time) plotted as a function of the order of the sounds (red, late window; black, onset window). (**F**) Average normalized responses to standard sounds in the onset window and offset window (red, late window; black, onset window. n=99). The left axis represents the scale for responses in the late window, and the right axis represents the scale for responses in the onset window. (**G**) PSTHs depicting the responses of the neuronal population to the 300 ms sounds, ranging from the first to the seventh order in the block (black, first; gray, second; blue, third; light blue, fourth; cyan, fifth; pink, sixth; red, seventh). The left axis represents the response scale for sounds from the second to the seventh, and the right axis represents the response scale for the first sound.

The online version of this article includes the following figure supplement(s) for figure 2:

**Figure supplement 1.** Comparison of neuronal responses between behavior protocol and non-behavior protocol.

**Figure supplement 2.** Response dynamic index (RDI) of different deviant sounds.

**Figure supplement 3.** Three-dimensional representation of recorded neurons based on response dynamic index (RDI).

Conversely, in the late window, responses ascended from the second to the fourth stimulus (ANOVA with post-hoc test: second versus fourth, p=8.29e-4, red line, **Figure 2F**) and maintained a stable level from the fourth to the seventh stimulus (p=0.653, ANOVA with post-hoc test, red line, **Figure 2F**). This pattern was mirrored in the peri-stimulus time histograms (PSTHs), which consolidated responses from the second to the seventh stimulus in one scale (left ordinate in **Figure 2G**) and the first stimulus in another scale (right ordinate in **Figure 2G**) for comparative purposes. Onset responses to the second through seventh stimuli completely overlapped, but responses in the late window diverged. The response profile remained flat in the late window for the first stimulus (black in **Figure 2G**), indicating a minimal climbing effect. Remarkably, starting from the second stimulus, the climbing effect began to manifest and progressively intensified, with a notable increase observed from the second to the fourth stimulus. This effect appeared to stabilize and overlap in responses from the fourth to the seventh sound (**Figure 2G**), illustrating the nuanced modulation of neuronal firing rates across successive sound presentations within the oddball sequence. The accumulation of the climbing effect alongside repetitive sound presentations suggests a potential linkage to reward prediction or sensory prediction, reflecting an increased probability of receiving a reward and the strengthening of sound prediction as the sound sequence progresses.

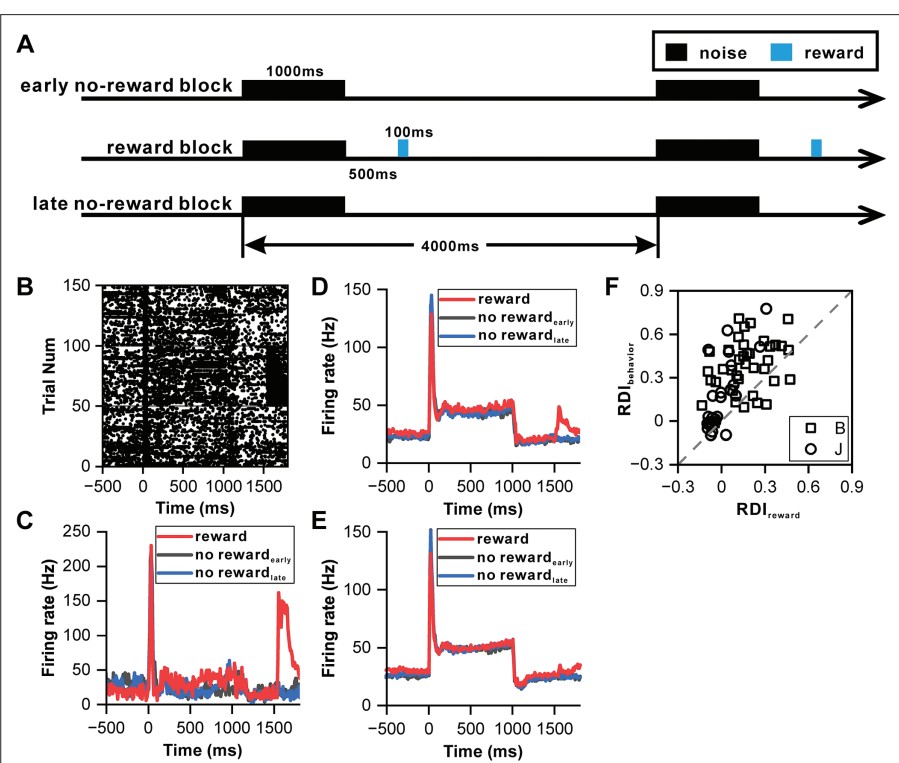

**Figure 3.** Influence of reward on inferior colliculus neurons. (**A**) Schematic representation of the reward protocol. The protocol involves the presentation of a 1000 ms white noise stimulus at 60 dB SPL, repeated 150 times with a 4 s interstimulus interval. The experimental design comprises three blocks: an initial 'early no-reward' block consisting of 50 trials without reward, a 'reward' block of 50 trials with water reward administered 500 ms after sound offset, and a final 'late no-reward' block of 50 trials without reward. (**B**) Raster plots of a representative neuron depicting the response patterns during the reward protocol. (**C**) PSTHs of the neuron in (**B**) representing the neuronal responses during the reward block, early no-reward block, and late no-reward block (red, reward block; black, early no-reward block; blue, late no-reward block). (**D**) PSTHs of the neuronal population exhibiting significant reward responses (n=24) representing the neuronal responses during the reward block, early no-reward block, and late no-reward block (red, reward block; black, early no-reward block; blue, late no-reward block). (**E**) Equivalent responses as in (**D**) but from neurons with nonsignificant reward responses (n=35). (**F**) Scatterplots of RDI in reward block versus RDI in behavior protocol (square, monkey B; circle, monkey J; n=63).

The online version of this article includes the following figure supplement(s) for figure 3:

**Figure supplement 1.** Influence of reward on neuronal responses.

## Reward effect on neuronal responses of IC neurons

To determine whether the observed climbing effect was driven by reward anticipation, we designed an experiment controlling for reward effects, thereby clarifying the underlying factors influencing IC neuronal activity. In this setup, a sequence of 1000 ms white noise bursts was played across 150 trials (*Figure 3A*), segmented into an initial 50-trial phase without reward (early no-reward block), a subsequent 50-trial phase with a water reward dispensed 500 ms after the sound offset (reward block), and a concluding 50-trial phase without reward (late no-reward block). An exemplary IC neuron exhibited sustained firing throughout the 1000 ms sound across all trials (*Figure 3B*), with a spike cluster coinciding with the reward delivery, indicating a reward-responsive behavior within the IC neuron (*Figure 3B*). The PSTHs for these conditions revealed nearly identical responses during the sound period (p=0.098, ANOVA with post-hoc test, *Figure 3C*), suggesting the absence of a climbing effect following the transient peak response.

Out of 59 IC neurons tested using this protocol, 24 demonstrated significant reward responses (p<0.05, paired t-test, *Figure 3D*), while 35 showed no significant reward response (*Figure 3E*). Across both neuron types, PSTHs following the transient peak response maintained a consistently flat profile (*Figure 3D and E*). However, the RDI during the reward condition, calculated from the normalized difference between responses in the late window and the after-peak window, indicated a behavior-dependent climbing effect, with most values lying above the unitary line (p=9.75e-11, paired t-test, *Figure 3F*). Furthermore, an additional set of 22 neurons was analyzed using 500 ms sounds within the reward protocol to ensure consistency in sound duration. This analysis showed that the PSTH firing rate increased in the behavioral condition but remained flat in the reward condition following the transient peak response (100–500 ms) (*Figure 3—figure supplement 1A*). The majority of these neurons' RDIs exceeded the unitary line in pairwise comparisons (p=3.02e-5, paired t-test, *Figure 3—figure supplement 1B*). These results demonstrate that reward anticipation does not drive the climbing effect, thereby reinforcing the idea that sensory prediction is the primary factor influencing the accumulation of the climbing effect in the IC.

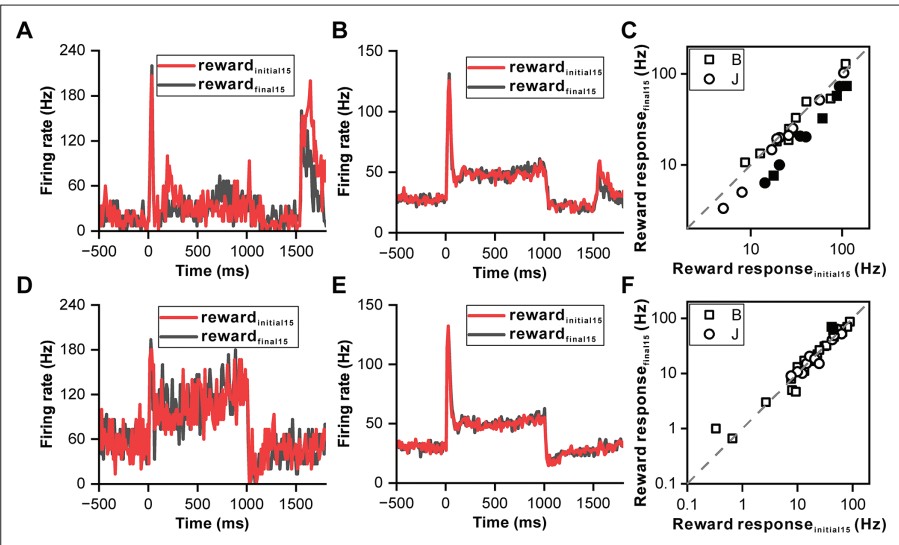

**Figure 4.** Reward prediction error in inferior colliculus neurons. (**A**) PSTHs of the representative neuron in *Figure 3B* depicting its response profiles during the initial 15 reward trials (red) and the final 15 reward trials (black). (**B**) PSTHs of the neuronal population with significant reward responses (n=28) demonstrating the firing activity of these neurons during the initial 15 reward trials (red) and the final 15 reward trials (black). (**C**) Scatterplots of reward responses in initial 15 reward trials versus final 15 reward trials from neurons in (**B**). The data points are color-coded, with solid circles representing neurons that exhibit a significant difference in reward responses between the initial and final reward trials. Conversely, open circles represent neurons wherein the difference in reward responses between the two trial sets is nonsignificant. The square and circle symbols correspond to monkey B and monkey J, respectively. (**D**) PSTHs of a neuron with nonsignificant reward responses demonstrating the response patterns of the neuron, similar to those presented in (**A**). (**E–F**) Equivalent PSTHs and scatterplots as in (**B–C**) but from neurons with nonsignificant reward responses (n=35).

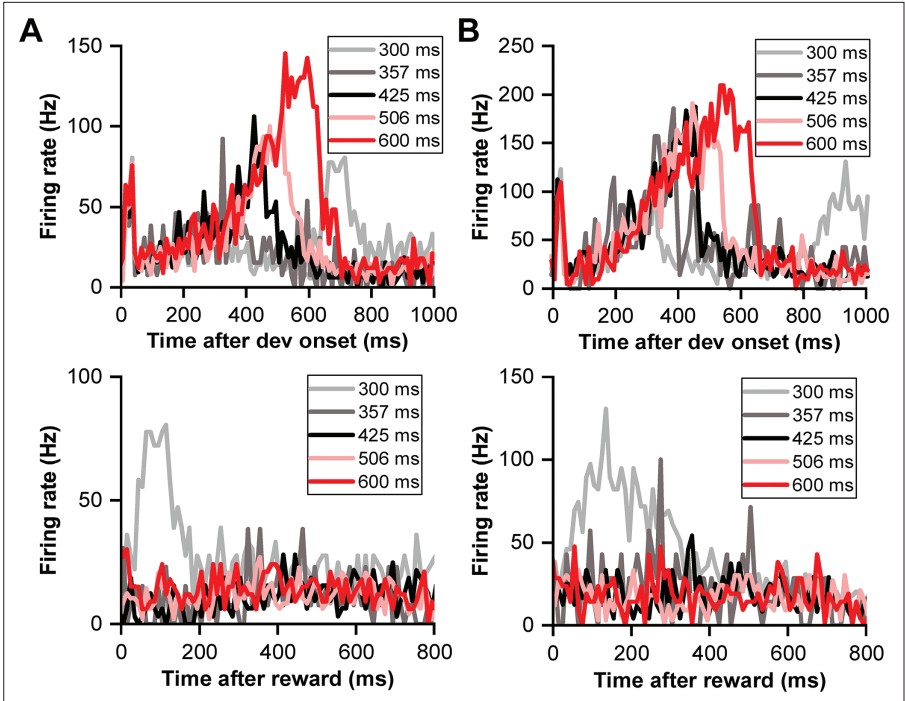

**Figure 5.** Reward prediction error in deviation detection behavior. (**A**) PSTHs illustrating neuronal responses aligned to deviant onset (top) and reward time (bottom). (**B**) PSTHs demonstrating neuronal responses aligned to deviant onset (top) and reward time (bottom) in another neuron.

The online version of this article includes the following figure supplement(s) for figure 5:

**Figure supplement 1.** Responses to the reward in deviation detection behavior.

## Reward prediction error in IC neuronal response

Recognizing that some IC neurons responded to reward delivery, we investigated whether these responses reflected reward prediction errors, thereby further elucidating the IC's role in reward processing. Notably, the reward response of the showcased neuron in *Figure 3B* was more pronounced at the start of the reward block. To explore this, we compared the first 15 trials to the last 15 trials within the reward block (*Figure 4A*), revealing that the reward response (0–200 ms relative to the reward onset) was significantly stronger during the initial trials (p=1.50e-2, ANOVA, *Figure 4A*). Twenty-eight neurons exhibiting significant reward responses were compiled; auditory responses were consistent across initial and final trials (p=0.534, ANOVA, *Figure 4B*), yet the reward responses in the initial trials were markedly higher (p=7.00e-3, ANOVA, *Figure 4B*). A comparison on an individual neuronal basis showed most points below the unity line (p=3.20e-3, paired t-test, *Figure 4C*), indicating a variance in reward responsiveness.

For another example neuron without a reward response (*Figure 4D*), auditory and reward responses between the first and last trials were indistinguishable for both phases (p=0.378 for auditory response, ANOVA, *Figure 4D*; p=0.637 for reward response period, ANOVA, *Figure 4D*). A collection of 35 neurons without reward responses exhibited similar firing rates for both auditory and reward response periods across trials (p=0.506 for auditory response, ANOVA, *Figure 4E*; p=0.585 for reward response period, ANOVA, *Figure 4E*), with the majority of pairwise comparisons aligning closely with the unity line (p=0.62, paired t-test, *Figure 4F*).

The heightened responses during the initial trials of the reward block suggest the presence of a reward prediction error, marked by an unexpected reward at the start, which gradually becomes predictable towards the block's end. This phenomenon was further supported by examining the responses in the duration deviation detection task. Since most IC neurons exhibit motor responses during key presses (*Figure 5—figure supplement 1*), which can complicate distinguishing between reward-related activity and motor responses, we specifically selected two neurons without motor responses during key presses (*Figure 5*). In *Figure 5*, two example IC neurons responded to the

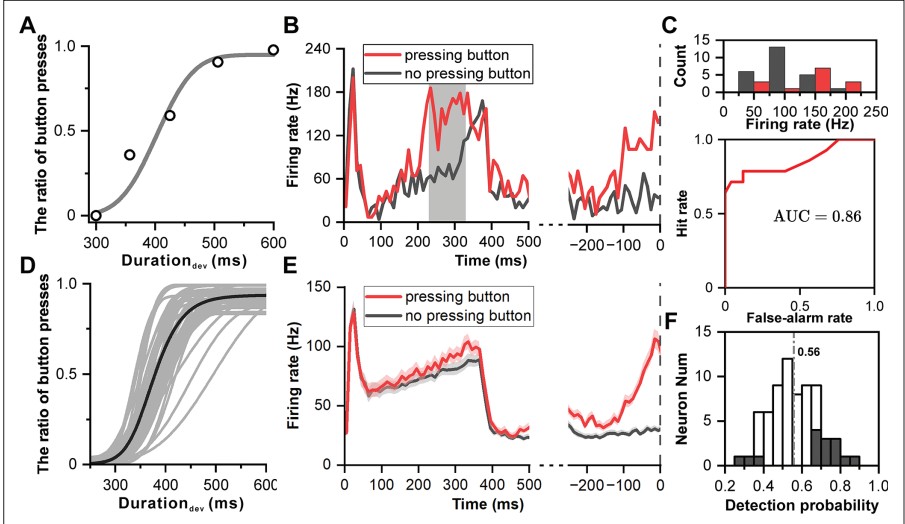

**Figure 6.** Neuronal responses of decision making in deviation detection behavior. (**A**) Cumulative Gaussian fits of psychophysical data in the example recording session. The duration of the deviant sound is plotted along the abscissa, while the ratio of button presses is plotted along the ordinate. (**B**) PSTHs showing example neuronal responses to deviant sound with 357 ms duration (red, pressing button after deviant sound; black, no pressing button after deviant sound). The left side of the subfigure shows the PSTHs aligned to the onset of the deviant sound, while the right side shows the PSTHs aligned to the time of button presses. (**C**) Top: Distribution of neuronal responses to 357 ms deviant sound categorized based on the behavioral choice made by the subject (red, pressing the button; black, no pressing the button). Bottom, receiver operating characteristic (ROC) analysis of comparison between two distributions. (**D**) Cumulative Gaussian fits to psychophysical data in population (n=69). (**E**) PSTHs showing neuronal responses to deviant sound with 357 ms duration in population (red, pressing button after deviant sound; black, no pressing button after deviant sound; left, aligned to deviant onset; right, aligned to time of pressing button). (**F**) Distribution of detection probability: This panel displays the distribution of detection probability (DP), with significant DP indicated by black bars.

The online version of this article includes the following figure supplement(s) for figure 6:

**Figure supplement 1.** Influence of sound numbers on neuronal responses to 357 ms deviant sound.

**Figure supplement 2.** Distribution of detection probability (DP) for 425 ms deviant sounds.

---

deviant sounds and subsequent rewards, each exhibiting a climbing effect post-peak response. The firing rate of these neurons returned to baseline post-sound for all durations, with a noticeable cluster of neuronal responses following sound presentation in the control condition, absent in all four deviant conditions (top row of *Figure 5*). When responses were aligned to reward delivery (the bottom row of *Figure 5*), notable reward responses were only observed in the control condition (p<0.001, ANOVA). The distinct response in the control condition, where the reward was unpredictable, contrasted sharply with the predictable reward scenario in the deviant condition, underscoring the ability of auditory IC neurons to encode reward prediction errors.

## Decision related signal of IC neurons in duration deviation detection

Finally, to determine whether the IC plays a role in decision-making processes related to auditory perception, we analyzed the correlation between neuronal activity and behavioral choices in the duration deviation detection task. During a session where a monkey excelled in performing the task (*Figure 6A*), trials were categorized based on the monkey's decisions—whether to press a button or not. Particularly in instances involving two specific deviant durations (357 ms and 425 ms), where the likelihood of pressing the button hovered around 50%, a sufficient number of trials were available to assess the correlation between neuronal activities and the monkey's decisions. This focus aims to delve deeper into how neuronal responses might inform or influence behavioral outcomes in these scenarios.

Analysis did not show the deviant response being influenced by the sequence of preceding standard sounds (*Figure 6—figure supplement 1*), leading to an aggregation of conditions for evaluation.

In an illustrative neuron responding to the 357 ms deviant sound, the activation was markedly stronger when the monkey identified the sound as deviant and opted to press the button (indicated by the red color in *Figure 6B*), in stark contrast to when the monkey did not perceive the sound as deviant, thus choosing not to press the button (illustrated in black color of *Figure 6B*). Aligning this observation with the moment of button press further accentuated the response disparity (p=1.32e-8, ANOVA, right panel in *Figure 6B*).

To quantitatively assess the decision related signal, an analysis incorporating Receiver Operating Characteristic (ROC) was conducted (*Figure 6C*), revealing a pronounced difference in firing rates aligned with the act of pressing the button (p=2.58e-6, ANOVA, *Figure 6C*). The area under the ROC curve (AUC), signifying detection probability, stood at 0.86 (p<0.001, permutation test), indicating a high likelihood that an ideal observer could discern between the two choices based solely on the neuronal firing rate.

Across 69 recorded sessions showcasing reliable behavioral outcomes (*Figure 6D*), population-level PSTHs highlighted a tendency for heightened responses during trials where the button was pressed (p=3.08e-4, ANOVA, left panel of *Figure 6E*), with a notable difference preceding the button press (p=7.93e-7, ANOVA, right panel of *Figure 6E*). The average detection probability across these sessions was 0.56, surpassing the chance level of 0.5 (p=2.57e-4, t-test, *Figure 6F*), with 15 out of 69 neurons showing significant detection probabilities. Specifically for the 425 ms sound trials, eight out of sixty-eight neurons displayed a significant detection probability, with an average value of 0.54, also above the chance level (p=4.30e-2, t-test, *Figure 6—figure supplement 2*).

## Discussion

The study unveils novel insights into the IC's role in auditory processing, highlighting its involvement beyond traditional sensory encoding. Neurons within the IC exhibit a sustained firing pattern, indicating a robust processing of sound duration, which is crucial for temporal perception (*Figure 1*). A behavior-dependent 'climbing effect' in neuronal firing rate suggests that IC's response to auditory stimuli is modulated by the context of the behavior, with a significant increase in response as the sound sequence progresses, implying a link to sensory experience (*Figure 2*). This effect was further supported by experiments that distinguished between behavior and reward conditions, showing that the climbing effect is contingent upon the behavioral context rather than mere reward anticipation (*Figure 2*, *Figure 3*). Additionally, IC neurons encoded reward prediction errors, demonstrating a nuanced capability to adjust responses based on reward predictability (*Figure 4*, *Figure 5*). Lastly, evidence of decision signals in IC neurons correlated with the monkey's decisions in a duration deviation detection task, indicating a direct involvement in decision-making processes related to auditory perception (*Figure 6*). Overall, our results strongly suggest that the IC is actively engaged in sensory experience, reward prediction and decision making, shedding light on its intricate functions in these processes.

### Sensory experience and response dynamics in IC

The IC neurons showed a rising firing rate after the onset peak (*Figure 2A–D*). The climbing effect was strongly modulated by perception (*Figure 2—figure supplement 1*), echoing results from previous research (*Metzger et al., 2006*). Metzger and colleagues reported a gradual increase in neural activity—termed late-trial ramping—in the IC during an auditory saccade task. Similar to our results, they observed no climbing effect in the absence of a behavioral task. Both studies support the idea that the climbing effect depends on both behavioral engagement and reward. While both pieces of research emphasize the IC's complex role in integrating auditory processing with cognitive functions related to reward and behavior, our findings provide further insight by distinguishing between the effects of sensory prediction and reward anticipation on IC neuronal activity. We demonstrated that the climbing effect is dynamically modulated (*Figure 2D–G*), and this modulation is driven primarily by sensory prediction rather than reward anticipation, as controlling for reward effects showed minimal impact on the response profile (*Figure 3D and E*). This modulation by preceding sensory experiences indicates that the IC is more than merely a relay station, suggesting a more intricate role in auditory processing influenced by both ascending and descending neural pathways. The IC's extensive descending network of connections, including those from the AC (*Adams, 1980*; *Andersen et al.,*

*1980*; *Bajo and Moore, 2005*; *Coleman and Clerici, 1987*; *Diamond et al., 1969*; *FitzPatrick and Imig, 1978*; *Winer et al., 2002*), thalamic nuclei (*Adams, 1980*; *Kuwabara and Zook, 2000*; *Winer et al., 2002*), and limbic system areas (*Adams, 1980*; *Coleman and Clerici, 1987*; *Larue et al., 2005*; *Marsh et al., 2002*), underscores its integration within broader sensory and cognitive processes, potentially underpinning the observed climbing effect.

The accumulation of the climbing effect across successive sound presentations appears to be intrinsically linked to the structure of oddball stimulation. Within the oddball paradigm, both sensory and reward predictions intensify alongside the recurrence of standard sounds, suggesting that the strength of these predictions could significantly influence neuronal responses. Our experimentation with rewards has effectively dismissed the role of reward prediction (*Figure 3*, *Figure 4*), highlighting the potential significance of sensory prediction in molding the climbing effect. Historically, oddball stimuli have been a focal point for investigating novelty detection within the AC (*Carbajal and Malmierca, 2018*; *Du et al., 2024*; *Gong et al., 2024*; *Gong et al., 2022*; *Malmierca et al., 2015*; *Rui et al., 2018*; *Song et al., 2023*; *Xu et al., 2017*; *Zhai et al., 2020*; *Zhai et al., 2019*), though many studies lacked incorporation of a relevant behavioral task (*Khouri and Nelken, 2015*; *Song et al., 2023*). Sensory prediction was indirectly assessed by contrasting predictive and non-predictive stimuli in control experiments, hence not addressing sensory prediction in a direct, real-time manner (*Carbajal and Malmierca, 2018*; *Fishman and Steinschneider, 2012*; *Parras et al., 2017*). The progression of the climbing effect with successive sound presentations could potentially offer a direct and real-time perspective on the impact of sensory prediction. Notably, while the onset response remained consistent from the second through the seventh sound (*Figure 2F*), significant dynamic changes in the late response window suggest a prolonged period is required for sensory prediction to exert its influence. Further research is required to explore the underlying neuronal mechanisms and functional significance of this dynamic change comprehensively.

## Decision related signal in IC

Having established how behavioral modulation affects responses in IC, we now delve into the relationship between IC responses and behavioral choices in a trial-by-trial analysis within a duration deviation detection task. This approach provides insight into the potential role of sensory neurons in perceptual tasks by comparing neural responses and perceptual decisions concurrently in the same subject. While numerous studies have explored this dynamic for cortical neurons (*Tsunada et al., 2016*), the contribution of subcortical neurons to decision-making processes remains less understood. To our knowledge, this study offers the direct correlation between IC neuron activity and perceptual choice, observed in real-time during experiments with animals (*Figure 1*). These correlations are crucial for understanding the mechanisms by which population encoding and decoding influence and direct behavior. Focusing on the deviation detection task—a ubiquitous aspect of sensory systems and an ongoing phenomenon in daily life—we employed an oddball paradigm with randomized presentations of standard sound numbers (*Figure 1*) to simulate natural occurrences of deviant stimuli. This approach allowed us to closely examine both the behavioral responses of monkeys and the corresponding neuronal correlates in the IC.

When neural responses were grouped based on behavioral choices, the PSTHs diverged (*Figure 6B*), indicating a strong correlation between neural firing variations and choice variations. Notably, within the overall population, the two PSTHs, sorted by the monkeys' choices, diverged before the end of the sound, converged after its end, and diverged once more prior to the movement decision (*Figure 6E*). This pattern during the sensory encoding phase underscores a potential critical role for IC in managing duration-sensitive behaviors. Traditionally, decision signals are thought to be predominantly driven by top-down inputs, as evidenced by previous studies (*Cumming and Nienborg, 2016*; *Nienborg et al., 2012*; *Nienborg and Cumming, 2009*). While the decision-related signal in the IC may derive from these top-down mechanisms, our findings also entertain the possibility that variations in IC neuronal activity could directly modulate higher brain areas, influencing behavioral outcomes. In either scenario, our findings clearly demonstrate the IC's significant involvement in cognitive processing, extending well beyond its traditional role as a mere auditory relay station.

Furthermore, neuronal responses to sound duration typically become more transient throughout the auditory pathway with neurons in the thalamus and cortex primarily showing onset responses (*Anderson and Linden, 2011*; deCharms and *deCharms and Merzenich, 1996*; *DeWeese et al.,*

*2003*; *He, 2001*; *Metzger et al., 2006*; *Nevue et al., 2015*; *Rauschecker et al., 1995*; *Xia et al., 2000*; *Yasui et al., 1991*; *Zhai et al., 2019*). However, exceptions exist, as sustained neurons have been identified in the medial geniculate body (MGB; *Allon et al., 1981*; *Bartlett and Wang, 2011*) and the AC (*Lu et al., 2001*; *Malone et al., 2002*; *Rauschecker et al., 1995*; *Wang et al., 2005*). In stark contrast, the majority of IC neurons exhibit sustained responses to prolonged sounds across various species, including frogs (*Ratnam and Feng, 1998*), bats (*Luo et al., 2008*), and rodents (*Pérez-González et al., 2006*; *Xia et al., 2000*; *Yin et al., 2008*). Our examination of IC neurons in macaque monkeys revealed a similar sustained firing pattern in response to long-duration sounds (*Figure 1A and B*), aligning with observations across diverse species. This evolutionary conservation of sustained response characteristics in the IC suggests its capacity to encode sound duration. However, the implications of this capability for behavior have remained uncertain. The present study offers direct evidence highlighting the significant role of the IC in decision-making processes related to duration perception (*Figure 6*).

## Reward effect and reward prediction error in IC

In addition to the role in sensory processing, another important finding in our study is the existence of a reward effect and a reward prediction error in the IC. The connection between the IC and the reward system is not very clear, but some studies have suggested that dopamine may play a role in modulating auditory processing in the IC (*Gittelman et al., 2013*; *Hoyt et al., 2019*; *Nevue et al., 2015*). Dopamine is a neurotransmitter that is involved in the reward system and other functions (*Hoyt et al., 2019*). Dopamine may have heterogeneous effects on the responses of many neurons in the IC, such as suppressing or enhancing neural activity, depending on various factors (*Hoyt et al., 2019*; *Nevue et al., 2015*). However, the exact mechanisms and functions of dopamine modulation in the IC are still not fully understood, particularly in primates. In this study, we first noticed that some auditory neurons responded to the reward (*Figure 3B and D*), but the reward modulation effect on auditory responses is slim in the awake monkey (*Figure 3B–E*).

Like the response of the dopaminergic neurons in the reward system (*Hollerman and Schultz, 1998*), the reward response of IC neurons displays prediction error signals, which were uncovered in two experiments. Firstly, during the experiment in which sound was followed by reward, the reward responses were significantly stronger in the initial 15 trials than in the final 15 trials (*Figure 4A–C*), as the reward is a surprise at the beginning while becoming a routine at the end. Secondly, during the deviation detection task, as the number of standard sounds was randomly chosen and the monkeys were not informed about the end of the stimulation in the control trials, the reward in such trials was not expected, but then a strong response was detected (*Figure 5*). By contrast, during the deviant trials, the monkey pressed the button, expected a reward, and no reward response was detected (*Figure 5*). Collectively, these findings affirm the presence of reward prediction error signals in the IC, underscoring its sophisticated engagement in reward-related auditory processing dynamics.

## Methods

### Key resources table

| Reagent type (species) or resource | Designation | Source or reference | Identifiers | Additional information |
|---|---|---|---|---|
| Biological sample (*Macaca mulatta*, male) | Monkey J;Monkey B | Hubei Topgene Biotechnology Co., Ltd. | http://en.topgenebio.com | |
| Software, algorithm | MATLAB R2022a | MathWorks | RRID:SCR_001622 | |
| Other | Auditory Workstation RZ6 | Tucker-Davis Technologies, TDT, Alachua, FL | https://www.tdt.com/component/rz6-multi-i-o-processor/ | |
| Other | Speaker, LS50 | KEF, UK | https://uk.kef.com/products/ls50-meta | |
| Other | ¼˝condenser microphone, 4954 | Brüel & Kjær, Nærum, Denmark | https://www.hbkworld.com/en | |
| Other | PHOTON/RT analyzer | Brüel & Kjær, Nærum, Denmark | https://www.hbkworld.com/en | |

*Continued on next page*

*Continued*

| Reagent type (species) or resource | Designation | Source or reference | Identifiers | Additional information |
|---|---|---|---|---|
| Other | Epoxy-coated tungsten microelectrodes | FHC Inc. | RRID:SCR_018426 | |
| Other | Remote-controlled microdrive | FHC Inc. | https://www.fh-co.com/product-category/star/ | |
| Other | 26-gauge transdural guide tubes | other | | Self-made by Yu lab. For more information, please contact yuxiongj@gmail.com. |
| Other | Plastic head-restraint ring | other | | Self-made by Yu lab. For more information, please contact yuxiongj@gmail.com. |
| Other | Recording grid | other | | Self-made by Yu lab. For more information, please contact yuxiongj@gmail.com. |

## Subjects and apparatus

Experiments were conducted in a sound-proof room. Acoustic stimuli were digitally generated using a computer-controlled Auditory Workstation (Tucker-Davis Technologies, TDT, Alachua, FL) and delivered via a loudspeaker (LS50, KEF, UK). The sound pressure was calibrated with a ¼″ condenser microphone (Brüel & Kjær 4954, Nærum, Denmark) and a PHOTON/RT analyzer (Brüel & Kjær). All stimuli were presented contralateral to the recording site.

Data were collected from two male rhesus monkeys (*M. mulatta*), which were chronically implanted with a plastic head-restraint ring and a recording grid, as described in detail in previous publications (*Gong et al., 2024*; *Yu et al., 2012*; *Yu et al., 2015*). All procedures were approved by the Animal Care and Use Committee of Zhejiang University (ZJU20200148) and were performed according to the National Institutes of Health Guide for the Care and Use of Laboratory Animals.

We recorded extracellular neural activity using epoxy-coated tungsten microelectrodes (FHC, 2–4 MΩ). Electrodes were inserted into the IC based on MRI structure (*Figure 1—figure supplement 1*) through 26-gauge transdural guide tubes and were advanced by a remote-controlled microdrive (FHC). Raw neural activity was amplified and filtered (300 Hz to 3000 Hz). The spike train of the neuron was analyzed off-line. Single units were identified based on waveform shape, latency, and amplitude. Only well-isolated neurons were included in the analyses.

## Auditory stimulation

Tones (100 ms; 5 ms rise-fall time) with multiple combinations of frequencies and intensities were presented to determine the frequency response areas (FRAs). Tones were presented randomly with 5 repetitions at each frequency (0.5–48 kHz in 26 logarithmic steps) and intensity (0–70 dB SPL in 10 dB steps) and interstimulus intervals of 300 ms. We used the FRAs to determine the characteristic frequency (CF).

To assess the ability of sustained firing in the IC, we presented white noise bursts with three kinds of duration (100 ms, 500 ms, 1000 ms) at 2 s interstimulus intervals. The three sound durations were randomly presented at 60 dB SPL, each for 20 trials.

To explore the reward effect on the neuronal firing of IC neurons, 1000 ms white noise was presented 150 times at 60 dB SPL at 4 s interstimulus interval (*Figure 3A*). The initial 50 trials consisted of no reward (early no-reward block), the middle 50 trials consisted of a reward of water 500 ms after the offset of each sound (reward block); and the final 50 trials again consisted of no reward (late no-reward block). (At the beginning of the experiment, we presented stimuli only 100 times, consisting of early no-reward block and reward block and collected 4 IC neurons.)

## Behavioral protocol

Animals were trained to perform a deviation detection task in two steps: (1) train the monkeys to press the button after the sound; (2) train the monkeys to press the button only after the deviant sound is presented. At both steps, the monkeys received a water reward when they made the correct response.

During the recording session, the sound was presented in blocks (*Figure 1C and D*). In each block, 3–6 repeated white noise bursts (standard sound) were presented at 600 ms interstimulus interval, and the last stimulus was either the same sound (the control condition) or a deviant sound (*Figure 1C*

*and D*). The monkey needed to press the button immediately within 600 ms after the offset of the deviant stimulus to get the reward of a drop of water. The reward was controlled electronically by a valve located outside the sound-proof room to prevent any noise interference from the valve. The task reaction time setting precluded the possibility that the monkey made the decision just because it was the last stimulus in the block; in the control condition, the monkey did not need to press the button within 600 ms after the onset of the last stimulus to get the reward. The number of standard stimuli was randomly selected from 3 to 6 to avoid that the monkey made the decision based on counting the number of sounds and avoided that the monkey guessed when the deviant would be presented, so that the monkey had to press the button based on the detection of the current deviant stimulus. In each recording session, the standard duration was kept the same (300 ms), but the deviant duration was systematically changed to control the difficulty of the task according to the following formula: $Duration_{deviant} = Duration_{standard} * \lambda^n$ (n=0,1,2,3,4) ($\lambda$ =1.19), where $Duration_{deviant}$ and $Duration_{standard}$ were deviant and standard duration, respectively. Thus, in each session, 5 conditions were presented. Each condition was usually presented 30–40 times. During the recording, all sounds were presented at 60 dB SPL with 5 ms rise-fall time in the profile.

We assessed the behavior of the monkey according to two criteria: (1) the monkey made the correct rejection. In the control condition, the ratio of pressing a button had to be less than 0.1; (2) The monkey made the correct hit. In the most differential condition, the ratio of pressing the button had to be greater than 0.85. Only the behavior meeting these two criteria was kept for further analysis.

## Data analysis

To characterize the climbing effect, we defined a factor termed the response dynamic index (RDI). RDI is calculated according to the following formula:

$$RDI = \frac{Fir_{offset}^{[-100\ 0]} - Fir_{onset}^{[100\ 200]}}{Fir_{offset}^{[-100\ 0]} + Fir_{onset}^{[100\ 200]}}$$

$Fir_{offset}^{[-100\ 0]}$ and $Fir_{onset}^{[100\ 200]}$ were the neuronal firing rates within the late window (−100–0 ms relative to the offset of the sound) and the after-peak window (100–200 ms relative to the onset of the sound), respectively. For quantitative descriptions of response dynamics, the onset window is defined as [0, 60] ms relative to the onset time, while the late window is defined as [–100, 0] ms relative to the offset time.

To define a reward-responsive neuron, we compared the response within 100 ms after the reward and the response within 200 ms before the reward was offered in the reward blocks (paired t-test). A neuron was considered reward-responsive only if the difference reached significance (p<0.05).

Behavioral performance was quantified by plotting the proportion of 'pressing button' choices as a function of duration difference between the deviant and standard sounds in ratio. Psychometric data were fit with a Gaussian integral function:

$$p\left(r\right) = \frac{1}{b\sqrt{2\pi}} \int_0^r e^{-\frac{(x-a)^2}{2b^2}}\ dx$$

In this expression, p is the proportion of pressing button choices, r is the ratio between deviant and standard duration.

## Resource availability

### Lead contact

Further information and requests for resources should be directed to and will be fulfilled by the lead contact, Xiongjie Yu (yuxiongj@gmail.com).

### Materials availability

This study did not generate new unique reagents.

## Acknowledgements

We are grateful to Prof. Liping Wang for their invaluable comments on the early version of the manuscript, as well as to Xiaokai Kou and Fujin Gao for their assistance with the experiments. This work was supported by STI2030-Major Projects (2022ZD0204800 and 2022ZD0204600) to XY, the National Natural Science Foundation of China (32171044 to XY and 32100827 to YZ).

## Additional information

### Funding

| Funder | Grant reference number | Author |
| --- | --- | --- |
| STI2030-Major Projects | 2022ZD0204800 | Xiongjie Yu |
| STI2030-Major Projects | 2022ZD0204600 | Xiongjie Yu |
| National Natural Science Foundation of China | 32171044 | Xiongjie Yu |
| National Natural Science Foundation of China | 32100827 | Yuying Zhai |

The funders had no role in study design, data collection and interpretation, or the decision to submit the work for publication.

### Author contributions
Xinyu Du, Data curation, Formal analysis, Investigation, Methodology, Software, Validation, Visualization, Writing – original draft, Writing – review and editing; Haoxuan Xu, Data curation, Formal analysis, Investigation, Methodology, Software, Validation, Visualization, Writing – review and editing; Peirun Song, Hangting Ye, Xuehui Bao, Qianyue Huang, Hisashi Tanigawa, Zhiyi Tu, Pei Chen, Xuan Zhao, Data curation, Investigation, Software; Yuying Zhai, Data curation, Funding acquisition, Investigation, Software; Josef P Rauschecker, Writing – original draft, Writing – review and editing; Xiongjie Yu, Conceptualization, Funding acquisition, Methodology, Project administration, Resources, Supervision, Writing – original draft

### Author ORCIDs
Xinyu Du  http://orcid.org/0009-0001-7773-1835
Haoxuan Xu  http://orcid.org/0000-0002-7316-8440
Hisashi Tanigawa  https://orcid.org/0000-0002-7736-0683
Xiongjie Yu  https://orcid.org/0000-0002-0040-2187

### Ethics
In this study, data were collected from two male rhesus monkeys (Macaca mulatta), which were chronically implanted with a plastic head-restraint ring and a recording grid, as described in detail in previous publications (Gong et al., 2024; Yu et al., 2012; Yu et al., 2015). All experimental procedures were conducted at Zhejiang University in Hangzhou, China. The procedures were approved by the Animal Care and Use Committee of Zhejiang University (ZJU20200148) and were performed in strict accordance with the principles outlined in the National Institutes of Health Guide for the Care and Use of Laboratory Animals. Throughout the study, the health and well-being of the monkeys were closely monitored daily by researchers and animal care staff. To enhance their welfare, enrichment activities were provided, including the introduction of toys and food-based rewards to encourage exploratory behavior within their home cages.

Reviewer #1 (Public review): https://doi.org/10.7554/eLife.101142.3.sa1
Reviewer #2 (Public review): https://doi.org/10.7554/eLife.101142.3.sa2
Author response https://doi.org/10.7554/eLife.101142.3.sa3

## Additional files

**Supplementary files**
MDAR checklist

**Data availability**
The raw data, preprocessed data, and the MATLAB code for the main content have been uploaded to Zenodo: https://doi.org/10.5281/zenodo.14539959.

The following dataset was generated:

| Author(s) | Year | Dataset title | Dataset URL | Database and Identifier |
|---|---|---|---|---|
| Du X, Xu H, Song P, Zhai Y, Ye H, Bao X, Huang Q, Tanigawa H, Tu Z, Chen P, Zhao X, Rauschecker JP, Yu X | 2024 | The Multifaceted Role of the Inferior Colliculus in Sensory Prediction, Reward Processing, and Decision-Making | https://doi.org/10.5281/zenodo.14539959 | Zenodo, 10.5281/zenodo.14539958 |

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
