## [Editor Report · eLife Assessment]

This **important** study presents a finding on the role of the Inferior Colliculus in sensory prediction, cognitive decision-making, and reward prediction. The evidence supporting the claims of the authors is **convincing**. The work will be of broad interest to sensory neuroscientists.

---

## [Referee Report · Reviewer #1 (Public review)]

Summary:

This work made a lot of efforts to explore the multifaceted roles of the inferior colliculus (IC) in auditory processing, extending beyond traditional sensory encoding. The authors recorded neuronal activity from the IC at single unit level when monkeys were passively exposed or actively engaged in behavioral task. They concluded that (1) IC neurons showed sustained firing patterns related to sound duration, indicating their roles in temporal perception, (2) IC neuronal firing rates increased as sound sequences progress, reflecting modulation by behavioral context rather than reward anticipation, (3) IC neurons encode reward prediction error and their capability of adjusting responses based on reward predictability, (4) IC neural activity correlates with decision-making. In summary, this study tried to provide a new perspective on IC functions by exploring its roles in sensory prediction and reward processing, what are not traditionally associated with this structure.

Strengths:

The major strength of this work is that the authors performed electrophysiological recordings from the IC of behaving monkeys. Compared with the auditory cortex and thalamus, the IC in monkeys has not been adequately explored.

Comments on revised version:

The authors have adequately addressed all my concerns.

---

## [Referee Report · Reviewer #2 (Public review)]

Summary:

The inferior colliculus (IC) has been explored for its possible functions in behavioral tasks and has been suggested to play more important roles rather than simple sensory transmission. The authors show us two major findings based on their experiments. The first one is climbing effect, which means that neurons' activities continue to increase along time course. The second one is reward effect, which refers to sudden increase of IC neurons' activities when the rewarding is given. Climbing effect is a surprising finding, but reward effect has not been explored clearly here.

Strengths:

Complex cognitive behaviors can be regarded as simple ideals of generating output based on information input, which depends on all kinds of input from sensory systems. The auditory system has hierarchic structures no less complex than those areas in charge of complex functions. Meanwhile, IC receives projections from higher areas, such as the auditory cortex, which implies IC is involved in complex behaviors. Experiments in behavioral monkeys are always time-consuming work with hardship, and this will offer more approximate knowledge of how the human brain works.

Weaknesses:

These findings are more about correlation but not causality of IC function in behaviors.

About 'reward effect', it is still unknown if the true nature of reward effect is the simple response to the sound elicited by the electromagnetic valve of rewarding system. The authors claimed the testing space is sound-proofed and believed this is enough to support their opinion. Since the electromagnetic valve was connected to the water tube, and the water tube was attached to a monkey-chair or even in monkey's mouth, the click sound may transmit to the monkey independently on air. There are simple ways to test what happens. One is to add a few trials without reward and see what happens, or to vary the latency between sound sequence and reward.

Only one of the major findings is convincing, this definitely reduces the credibility of the authors' statements.

---

## [Author Response]

The following is the authors' response to the original reviews.

**Public Reviews:**

**Reviewer #1 (Public review):**
Summary:This work made a lot of efforts to explore the multifaceted roles of the inferior colliculus (IC) in auditory processing, extending beyond traditional sensory encoding. The authors recorded neuronal activitity from the IC at single unit level when monkeys were passively exposed or actively engaged in behavioral task. They concluded that (1) IC neurons showed sustained firing patterns related to sound duration, indicating their roles in temporal perception, (2) IC neuronal firing rates increased as sound sequences progress, reflecting modulation by behavioral context rather than reward anticipation, (3) IC neurons encode reward prediction error and their capability of adjusting responses based on reward predictability, (4) IC neural activity correlates with decision-making. In summary, this study tried to provide a new perspective on IC functions by exploring its roles in sensory prediction and reward processing, which are not traditionally associated with this structure.Strengths:The major strength of this work is that the authors performed electrophysiological recordings from the IC of behaving monkeys. Compared with the auditory cortex and thalamus, the IC in monkeys has not been adequately explored.

We appreciate the reviewer's acknowledgment of the efforts and strengths of our study. Indeed, our goal was to provide a comprehensive exploration of the multifaceted roles of the inferior colliculus (IC) in auditory processing and beyond, particularly in sensory prediction and reward processing. The use of electrophysiological recordings in behaving monkeys was central to our approach, as we sought to uncover the underexplored aspects of IC function in these complex cognitive domains. We are pleased that the reviewer recognizes the value of investigating the IC, a structure that has not been adequately explored in primates compared to other auditory regions like the cortex and thalamus. This feedback reinforces our belief that our work contributes significantly to advancing the understanding of the IC's roles in cognitive processing.

We look forward to addressing any further points the reviewers may have and refining our manuscript accordingly. Thank you for your constructive feedback and for recognizing the strengths of our research approach.

Weaknesses:(1) The authors cited several papers focusing on dopaminergic inputs in the IC to suggest the involvement of this brain region in cognitive functions. However, all those cited work were done in rodents. Whether monkey's IC shares similar inputs is not clear.

We appreciate the reviewer's insightful comment on the limitations of extrapolating findings from rodent models to monkeys, particularly concerning dopaminergic inputs to the Inferior Colliculus (IC). While it is true that most studies on dopaminergic inputs to the IC have been conducted in rodents, to our knowledge, no studies have been conducted specifically in primates. To address the reviewer's concern, we have added a statement in both the introduction and discussion sections of our manuscript:

Introduction: "However, these studies were conducted in rodents, and the existence and role of dopaminergic inputs in the primate IC remain underexplored." (P.5, Line. 16-17)Discussion: "However, the exact mechanisms and functions of dopamine modulation in the inferior colliculus are still not fully understood, particularly in primates. " (P.21, Line. 7-9)

(2) The authors confused the two terms, novelty and deviation. According to their behavioral paradigm, deviation rather than novelty should be used in the paper because all the stimuli have been presented to the monkeys during training. Therefore, there is actually no novel stimuli but only deviant stimuli. This reflects that the author has misunderstood the basic concept.

We appreciate the reviewer's clarification regarding the distinction between "novelty" and "deviation" in the context of our behavioral paradigm. We agree that, given the nature of our experimental design where all stimuli were familiar to the monkeys during training, the term "deviation" more accurately describes the stimuli used in our study rather than "novelty."

To address this, we have revised the manuscript to replace the term "novelty" with "deviation" wherever applicable. This change has been made to ensure accurate terminology is used throughout the paper, thereby eliminating any potential misunderstanding of the concepts involved in our study.

We thank the reviewer for pointing out this important distinction, which has improved the clarity and precision of our manuscript.

(3) Most of the conclusions were made based on correlational analysis or speculation without providing causal evidences.

We appreciate the reviewer's concern regarding the reliance on correlational analyses in our study. Indeed, we acknowledge that the conclusions drawn primarily reflect correlations between neuronal activity and behavioral outcomes, rather than direct causal evidence. This limitation is common in many electrophysiological studies, particularly those conducted in behaving primates, where directly manipulating specific neural circuits to establish causality presents significant challenges, especially in comparison to research in mice.

This complexity is further compounded when considering the IC's role as a key lower-level relay station in the auditory pathway. Manipulating IC activity could have a widespread impact on auditory responses in downstream pathways, potentially influencing sensory prediction and decision-making processes.

Despite this limitation, our study provides novel evidence suggesting that the IC may exhibit multiple facets of cognitive signaling, which could inspire future research aimed at exploring the underlying mechanisms and broader functional implications of these signals.

To address the reviewer's concerns, we have made the following adjustments to the manuscript:

(1) Clarified the Scope of Conclusions: We have revised the language in the Results and Discussion sections to explicitly state that our findings represent correlational relationships rather than causal mechanisms. For example, we have referred to the associations observed between IC activity and behavioral outcomes as "correlational" and have refrained from making definitive causal claims without supporting experimental evidence.

"Finally, to determine whether the IC plays a role in decision-making processes related to auditory perception, we analyzed the correlation between neuronal activity and behavioral choices in the duration deviation detection task." (P.14, Line. 4-6)

(2) Proposed Future Directions: In the Discussion section, we have included suggestions for future studies to directly test the causality of the observed relationships.

"Further research is required to explore the underlying neuronal mechanisms and functional significance of this dynamic change comprehensively." (P.18, Line. 11-12)

We believe these revisions provide a more balanced interpretation of our findings while emphasizing the importance of future research to build on our results and establish causal relationships. Thank you for raising this critical point, which has led to a more rigorous and transparent presentation of our study.

(4) Results are presented in a very "straightforward" manner with too many detailed descriptions of phenomena but lack of summary and information synthesis. For example, the first section of Results is very long but did not convey clear information.

We appreciate the reviewer's feedback regarding the presentation of our results. We understand that the detailed descriptions of phenomena may have made it difficult to discern the key findings and overarching themes in the study. We recognize the importance of balancing detailed reporting with clear summaries and synthesis to effectively communicate our findings.

To address this concern, we have made the following revisions to the manuscript:

(1) Condensed and Synthesized Key Findings: We have streamlined the presentation of the Results section by condensing overly detailed descriptions and focusing on the most critical aspects of the data. Key findings are now summarized at the end of each subsection to ensure that the main points are clearly conveyed.

"The accumulation of the climbing effect alongside repetitive sound presentations suggests a potential linkage to reward prediction or sensory prediction, reflecting an increased probability of receiving a reward and the strengthening of sound prediction as the sound sequence progresses." (P.10, Line. 17-20)

"The distinct response in the control condition, where the reward was unpredictable, contrasted sharply with the predictable reward scenario in the deviant condition, underscoring the ability of auditory IC neurons to encode reward prediction errors." (P.13, Line. 21-22; P.14, Line. 1-2)

(2) Improved Flow and Clarity: We have revised the structure and organization of the Results section to improve the flow of information. By rearranging certain paragraphs and refining the language, we aim to present the results in a more cohesive and coherent manner.

"Deviant Response dynamics in duration deviation detection" (P.6, Line. 12)

"Standard Response dynamics in duration deviation detection" (P.9, Line. 4)

We believe these changes will make the Results section more accessible and informative, allowing readers to more easily grasp the significance of our findings. Thank you for your valuable suggestion, which has significantly improved the clarity and impact of our manuscript.

(5) The logic between different sections of Results is not clear.

We appreciate the reviewer's observation regarding the lack of clear logical connections between different sections of the Results. We acknowledge that a coherent flow is essential for effectively communicating the progression of findings and their implications.

To address this concern, we have made the following revisions:

(1) Enhanced Transitions Between Sections: We have introduced clearer transitional statements between sections of the Results. These transitions explicitly state how each new section builds upon or relates to the previous findings, creating a more cohesive narrative.

"Building upon the findings from the deviant responses, we next explored whether the climbing effect also manifested in responses to preceding standard stimuli, thereby examining the influence of sensory prediction and repetition on IC neuronal activity." (P.9, Line. 5-7)

"To determine whether the observed climbing effect was driven by reward anticipation, we designed an experiment controlling for reward effects, thereby clarifying the underlying factors influencing IC neuronal activity." (P.10, Line. 22; P.11, Line. 1-2)

"Recognizing that some IC neurons responded to reward delivery, we investigated whether these responses reflected reward prediction errors, thereby further elucidating the IC's role in reward processing." (P.12, Line. 9-11)

"Finally, to determine whether the IC plays a role in decision-making processes related to auditory perception, we analyzed the correlation between neuronal activity and behavioral choices in the duration deviation detection task." (P.14, Line. 4-6)

(2) Integration of Findings: In several places within the Results, we have added brief synthesis paragraphs that integrate findings across sections. These integrative summaries help to tie together the different aspects of our study, demonstrating how they collectively contribute to our understanding of the Inferior Colliculus's (IC) role in sensory prediction, decision-making, and reward processing.

"These results demonstrate that reward anticipation does not drive the climbing effect, thereby reinforcing the idea that sensory prediction is the primary factor influencing the accumulation of the climbing effect in the IC." (P.12, Line. 4-7)

"The distinct response in the control condition, where the reward was unpredictable, contrasted sharply with the predictable reward scenario in the deviant condition, underscoring the ability of auditory IC neurons to encode reward prediction errors." (P.13, Line. 21-22; P.14, Line. 1-2)

(3) Clarified Rationale: At the beginning of each major section, we have clarified the rationale behind why certain experiments were conducted, connecting them more clearly to the overarching goals of the study. This should help the reader understand the purpose of each set of results in the context of the broader research objectives.

"Building upon the findings from the deviant responses, we next explored whether the climbing effect also manifested in responses to preceding standard stimuli, thereby examining the influence of sensory prediction and repetition on IC neuronal activity." (P.9, Line. 5-7)

"To determine whether the observed climbing effect was driven by reward anticipation, we designed an experiment controlling for reward effects, thereby clarifying the underlying factors influencing IC neuronal activity." (P.10, Line. 22; P.11, Line. 1-2)

"Recognizing that some IC neurons responded to reward delivery, we investigated whether these responses reflected reward prediction errors, thereby further elucidating the IC's role in reward processing." (P.12, Line. 9-11)

"Finally, to determine whether the IC plays a role in decision-making processes related to auditory perception, we analyzed the correlation between neuronal activity and behavioral choices in the duration deviation detection task." (P.14, Line. 4-6)

We believe these changes improve the overall coherence and readability of the Results section, allowing readers to better follow the logical progression of our study. We are grateful for this constructive feedback and believe it has significantly enhanced the manuscript.

(6) In the Discussion, there is excessive repetition of results, and further comparison with and discussion of potentially related work are very insufficient. For example, Metzger, R.R., et al. (J Neurosc, 2006) have shown similar firing patterns of IC neurons and correlated their findings with reward.

We appreciate the reviewer's insightful critique regarding the excessive repetition in the Discussion and the lack of sufficient comparison with related work. We acknowledge that a well-balanced Discussion should not only interpret findings but also place them in the context of existing literature to highlight the novelty and significance of the study.

To address these concerns, we have made the following revisions:

(1) Reduction of Repetition: We have carefully revised the Discussion to minimize redundant repetition of the Results. Instead of restating the findings, we now focus more on their implications, limitations, and how they advance the current understanding of the Inferior Colliculus (IC) and its broader cognitive roles.

"We demonstrated that the climbing effect is dynamically modulated (Figure 2D-G), and this modulation is driven primarily by sensory prediction rather than reward anticipation, as controlling for reward effects showed minimal impact on the response profile (Figure 3D, E). This modulation by preceding sensory experiences indicates that the IC is more than merely a relay station, suggesting a more intricate role in auditory processing influenced by both ascending and descending neural pathways." (P.17, Line. 1-5)

(2) Incorporation of Related Work: We have expanded the Discussion to include a more comprehensive comparison with existing literature, specifically highlighting studies that have reported similar findings. For example, we now discuss the work by Metzger et al. (2006), which demonstrated similar firing patterns of IC neurons and correlated these with reward-related processes. This comparison helps contextualize our results and emphasizes the novel contributions our study makes to the field.

"Metzger and colleagues reported a gradual increase in neural activity—termed late-trial ramping—in the IC during an auditory saccade task. Similar to our results, they observed no climbing effect in the absence of a behavioral task. Both studies support the idea that the climbing effect depends on both behavioral engagement and reward. While both pieces of research emphasize the IC's complex role in integrating auditory processing with cognitive functions related to reward and behavior, our findings provide further insight by distinguishing between the effects of sensory prediction and reward anticipation on IC neuronal activity." (P.16, Line. 16-24)

We believe these revisions have significantly improved the quality of the Discussion by reducing unnecessary repetition and providing a more thorough engagement with the relevant literature. We are grateful for the reviewer's valuable feedback, which has helped us refine and strengthen the manuscript.

**Reviewer #2 (Public review):**
Summary:The inferior colliculus (IC) has been explored for its possible functions in behavioral tasks and has been suggested to play more important roles rather than simple sensory transmission. The authors revealed the climbing effect of neurons in IC during decision-making tasks, and tried to explore the reward effect in this condition.Strengths:Complex cognitive behaviors can be regarded as simple ideals of generating output based on information input, which depends on all kinds of input from sensory systems. The auditory system has hierarchic structures no less complex than those areas in charge of complex functions. Meanwhile, IC receives projections from higher areas, such as auditory cortex, which implies IC is involved in complex behaviors. Experiments in behavioral monkeys are always time-consuming works with hardship, and this will offer more approximate knowledge of how the human brain works.

We greatly appreciate the reviewer's positive summary of our work and recognition of the effort involved in conducting experiments on behaving monkeys. We agree with the reviewer that the inferior colliculus (IC) plays a significant role beyond mere sensory transmission, particularly in integrating sensory inputs with higher cognitive functions. Our study aims to shed light on these complex functions by revealing the climbing effect of IC neurons during decision-making tasks and exploring how reward influences this dynamic.

We are encouraged that the reviewer acknowledges the importance of investigating the IC's role within the broader framework of complex cognitive behaviors and appreciates the hierarchical nature of the auditory system. The reviewer's comments reinforce the value of our research in contributing to a more nuanced understanding of how the IC might contribute to sensory-cognitive integration.

We thank the reviewer for highlighting the significance of using behavioral monkey models to approximate human brain function. We are hopeful that our findings will serve as a stepping stone for further research exploring the multifaceted roles of the IC in cognition and behavior.

We will now proceed to address the specific concerns and suggestions provided by the reviewer in the following sections.

Weaknesses:These findings are more about correlation but not causality of IC function in behaviors. And I have a few major concerns.

We appreciate the reviewer's concern regarding the reliance on correlational analyses in our study. We fully acknowledge the importance of distinguishing between correlation and causality. As outlined in our response to Question 3 from Reviewer #1, we recognize the limitations of relying on correlational data and the inherent challenges in establishing direct causal links, particularly in electrophysiological studies involving behaving primates, and given the lower-level role of the IC in the auditory pathway.

We have taken steps to clarify this distinction throughout our manuscript. Specifically, we have revised the Results and Discussion sections to ensure that the findings are presented as correlational, not causal, and we have proposed future studies utilizing more direct manipulation techniques to assess causality. We hope these revisions adequately address your concerns.

"Finally, to determine whether the IC plays a role in decision-making processes related to auditory perception, we analyzed the correlation between neuronal activity and behavioral choices in the duration deviation detection task." (P.14, Line. 4-6)

"Further research is required to explore the underlying neuronal mechanisms and functional significance of this dynamic change comprehensively." (P.18, Line. 11-12)

Comparing neurons' spike activities in different tests, a 'climbing effect' was found in the oddball paradigm. The effect is clearly related to training and learning process, but it still requires more exploration to rule out a few explanations. First, repeated white noise bursts with fixed inter-stimulus-interval of 0.6 seconds was presented, so that monkeys might remember the sounds by rhymes, which is some sort of learned auditory response. It is interesting to know monkeys' responses and neurons' activities if the inter-stimuli-interval is variable. Second, the task only asked monkeys to press one button and the reward ratio (the ratio of correct response trials) was around 78% (based on the number from Line 302). so that, in the sessions with reward, monkeys had highly expected reward chances, does this expectation cause the climbing effect?

We thank the reviewer for raising these insightful points regarding the 'climbing effect' observed in the oddball paradigm and its potential relationship with training, learning processes, and reward expectation. Below, we address each of the reviewer's specific concerns:

(1) Inter-Stimulus Interval (ISI) and Rhythmic Auditory Response:

The reviewer suggests that the fixed inter-stimulus interval (ISI) of 0.6 seconds might lead to a rhythmic auditory response, where monkeys could anticipate the sounds. We appreciate this perspective and recognize its relevance. However, we believe that rhythm is unlikely to be a significant contributor to the 'climbing effect' for two key reasons:

a) The 'climbing effect' begins as early as the second sound in the block (as shown in Fig. 2D and Fig. 3B), before any rhythm or pattern could be fully established, since rhythm generally requires at least three repetitions to form.

b) In our reward experiment (Figs. 4-5), the sounds were also presented at regular ISIs, which could have facilitated rhythmic learning, yet the observed climbing effect was comparatively small in those conditions.

Unfortunately, we did not explore variable ISIs in this current study, so we cannot directly address this concern with the available data.

(2) Reward Expectation and Climbing Effect:

The reviewer raises a valid concern regarding whether the 'climbing effect' might be influenced by the monkeys' high reward expectation, especially given the high reward ratio (~78%) in the sessions. While it is plausible that reward expectation could contribute to the observed increase in neuronal firing rates, we believe the results from our reward experiment (Fig. 4) suggest otherwise.

In this experiment, even though reward expectation was likely formed due to the consistent pairing of sounds with rewards (100% reward delivery), we did not observe a significant climbing effect in the auditory response. Additionally, the presence of reward prediction error (Fig. 4D) further supports the idea that while the monkeys may indeed form reward expectations, these expectations do not directly drive the climbing effect in the IC.

To make this distinction clearer, we have added sentences in the revised manuscript explicitly discussing the relationship between reward expectation and the climbing effect.

"Within the oddball paradigm, both sensory and reward predictions intensify alongside the recurrence of standard sounds, suggesting that the strength of these predictions could significantly influence neuronal responses. Our experimentation with rewards has effectively dismissed the role of reward prediction (Figures 3 and 4), highlighting the potential significance of sensory prediction in molding the climbing effect." (P.17, Line. 14-19)

We believe these revisions provide a clearer understanding of the factors contributing to the climbing effect and effectively address the reviewer's concerns. We sincerely thank the reviewer for these valuable suggestions, which have allowed us to improve the clarity and depth of our manuscript.

"Reward effect" on IC neurons' responses were shown in Fig. 4. Is this auditory response caused by physical reward action or not? In reward sessions, IC neurons have obvious response related to the onset of water reward. The electromagnetic valve is often used in water-rewarding system and will give out a loud click sound every time when the reward is triggered. IC neurons' responses may be simply caused by the click sound if the electromagnetic valve is used. It is important to find a way to rule out this simple possibility.

We appreciate the reviewer's concern regarding the potential confounding factor introduced by the electromagnetic valve's click sound during water reward delivery, which could be misinterpreted as an auditory response rather than a response to the reward itself. Anticipating this possibility, we took measures to eliminate it by placing the electromagnetic valve outside the soundproof room where the neuronal recordings were performed.

To address your concern more explicitly, we have added sentences in the Methods section of the revised manuscript detailing this setup, ensuring that readers are aware of the steps we took to eliminate this potential confound. By doing so, we believe that the observed reward-related neural activity in the IC is attributable to the reward processing itself rather than an auditory response to the valve click. We appreciate you bringing this important aspect to our attention, and we hope our clarification strengthens the interpretation of our findings.

"The reward was controlled electronically by a valve located outside the sound-proof room to prevent any noise interference from the valve." (P.24, Line. 6-7)

**Reviewer #3 (Public review):**
Summary:The authors aimed to investigate the multifaceted roles of the Inferior Colliculus (IC) in auditory and cognitive processes in monkeys. Through extracellular recordings during a sound duration-based novelty detection task, the authors observed a "climbing effect" in neuronal firing rates, suggesting an enhanced response during sensory prediction. Observations of reward prediction errors within the IC further highlight its complex integration in both auditory and reward processing. Additionally, the study indicated IC neuronal activities could be involved in decision-making processes.Strengths:This study has the potential to significantly impact the field by challenging the traditional view of the IC as merely an auditory relay station and proposing a more integrative role in cognitive processing. The results provide valuable insights into the complex roles of the IC, particularly in sensory and cognitive integration, and could inspire further research into the cognitive functions of the IC.

We appreciate the reviewer's positive summary of our work and recognition of its potential impact on the field. We are pleased that the reviewer acknowledges the significance of our findings in challenging the traditional view of the Inferior Colliculus (IC) as merely an auditory relay station and in proposing its integrative role in cognitive processing.

Our study indeed aims to provide new insights into the multifaceted roles of the IC, particularly in the context of sensory and cognitive integration. We believe that this research could pave the way for future studies that further explore the cognitive functions of the IC and its involvement in complex behavioral processes.

We are encouraged by the reviewer's positive assessment and are committed to continuing to refine our work in response to the constructive feedback provided. We hope that our findings will contribute to advancing the understanding of the IC's role in the broader context of neuroscience.

We will now proceed to address the specific concerns and suggestions provided by the reviewer in the following sections.

Weaknesses:Major Comments:(1) Structural Clarity and Logic Flow:The manuscript investigates three intriguing functions of IC neurons: sensory prediction, reward prediction, and cognitive decision-making, each of which is a compelling topic. However, the logical flow of the manuscript is not clearly presented and needs to be well recognized. For instance, Figure 3 should be merged into Figure 2 to present population responses to the order of sounds, thereby focusing on sensory prediction. Given the current arrangement of results and figures, the title could be more aptly phrased as "Beyond Auditory Relay: Dissecting the Inferior Colliculus's Role in Sensory Prediction, Reward Prediction, and Cognitive Decision-Making."

We appreciate the reviewer's detailed feedback on the structural clarity and logical flow of the manuscript. We understand the importance of presenting our findings in a clear and cohesive manner, especially when addressing multiple complex topics such as sensory prediction, reward prediction, and cognitive decision-making.

To address the reviewer's concerns, we have made the following revisions:

(1) Reorganization of Figures and Results:

We agree with the suggestion to merge Figure 3 into Figure 2. By doing so, we can present the population responses to the order of sounds more effectively, thereby streamlining the focus on sensory prediction. This will allow readers to more easily follow the progression of the results related to this key function of the IC.

We have reorganized the Results section to ensure a smoother transition between the different aspects of IC function that we are investigating. The new structure will better guide the reader through the narrative, aligning with the themes of sensory prediction, reward prediction, and cognitive decision-making.

"Deviant Response dynamics in duration deviation detection" (P.6, Line. 12)

"Standard Response dynamics in duration deviation detection" (P.9, Line. 4)

(2) Revised Title:

In line with the reviewer's suggestion, we have revised the title to "Beyond Auditory Relay: Dissecting the Inferior Colliculus's Role in Sensory Prediction, Reward Prediction, and Cognitive Decision-Making." We believe this title more accurately reflects the scope and focus of our study, as it highlights the three core functions of the IC that we are investigating.

(3) Improved Logic Flow:

We have added introductory statements at the beginning of each section within the Results to clarify the rationale behind the experiments and the logical connections between them. This should help to improve the overall flow of the manuscript and make the progression of our findings more intuitive for readers.

"Building upon the findings from the deviant responses, we next explored whether the climbing effect also manifested in responses to preceding standard stimuli, thereby examining the influence of sensory prediction and repetition on IC neuronal activity." (P.9, Line. 5-7)

"To determine whether the observed climbing effect was driven by reward anticipation, we designed an experiment controlling for reward effects, thereby clarifying the underlying factors influencing IC neuronal activity." (P.10, Line 22; P.11, Line. 1-2)

"Recognizing that some IC neurons responded to reward delivery, we investigated whether these responses reflected reward prediction errors, thereby further elucidating the IC's role in reward processing." (P.12, Line. 9-11)

"Finally, to determine whether the IC plays a role in decision-making processes related to auditory perception, we analyzed the correlation between neuronal activity and behavioral choices in the duration deviation detection task." (P.14, Line. 4-6)

We believe these changes significantly enhance the clarity and logical structure of the manuscript, making it easier for readers to understand the sequence and importance of our findings. Thank you for your valuable suggestion, which has led to a more coherent and focused presentation of our work.

(2) Clarification of Data Analysis:Key information regarding data analysis is dispersed throughout the results section, which can lead to confusion. Providing a more detailed and cohesive explanation of the experimental design would significantly enhance the interpretation of the findings. For instance, including a detailed timeline and reward information for the behavioral paradigms shown in Figures 1C and D would offer crucial context for the study. More importantly, clearly presenting the analysis temporal windows and providing comprehensive statistical analysis details would greatly improve reader comprehension.

We appreciate the reviewer's insightful comment regarding the need for clearer and more cohesive explanations of the data analysis and experimental design. We recognize that a well-structured presentation of this information is essential for the reader to fully understand and interpret our findings. To address this, we have made the following revisions:

(1) Detailed Explanation of Experimental Design:

We have included a more detailed explanation of the experimental design, particularly for the behavioral paradigms shown in Figures 1C and 1D. This includes a comprehensive timeline of the experiments, along with explicit information about the reward structure and timing. By providing this context upfront, we aim to give readers a clearer understanding of the conditions under which the neuronal recordings were obtained.

(2) Cohesive Presentation of Data Analysis:

Key information regarding data analysis, which was previously dispersed throughout the Results section, has been consolidated and moved to a dedicated subsection within the Methods. This subsection now provides a step-by-step description of the analysis process, including the temporal windows used for examining neuronal activity, as well as the specific statistical methods employed.

We have also ensured that the temporal windows used for different analyses (e.g., onset window, late window, etc.) are clearly defined and consistently referenced throughout the manuscript. This will help readers track the use of these windows across different figures and analyses.

(3) Enhanced Statistical Analysis Details:

We have expanded the description of the statistical analyses performed in the study, including the rationale behind the choice of tests, the criteria for significance, and any corrections for multiple comparisons. This relevant information is highlighted in the Results section or figure legends to facilitate understanding.

We believe these changes will significantly improve the clarity and comprehensibility of the manuscript, allowing readers to better follow the experimental design, data analysis, and the conclusions drawn from our findings. Thank you for this valuable feedback, which has helped us to enhance the rigor and transparency of our presentation.

(3) Reward Prediction Analysis:The conclusion regarding the IC's role in reward prediction is underdeveloped. While the manuscript presents evidence that IC neurons can encode reward prediction, this is only demonstrated with two example neurons in Figure 6. A more comprehensive analysis of the relationship between IC neuronal activity and reward prediction is necessary. Providing population-level data would significantly strengthen the findings concerning the IC's complex functionalities. Additionally, the discussion of reward prediction in lines 437-445, which describes IC neuron responses in control experiments, does not sufficiently demonstrate that IC neurons can encode reward expectations. It would be valuable to include the responses of IC neurons during trials with incorrect key presses or no key presses to better illustrate this point.

We deeply appreciate the detailed feedback provided regarding the conclusions on the inferior colliculus (IC)'s role in reward prediction within our manuscript. We acknowledge the importance of a robust and comprehensive presentation of our findings, particularly when discussing complex neural functionalities.

In response to the reviewers' concerns, we have made the following revisions to strengthen our manuscript:

(1) Inclusion of Population-Level Data for IC Neurons:

In the revised manuscript, we have included population-level results for IC neurons in a supplementary figure. Initially, we focused on two example neurons that did not exhibit motor-related responses to key presses to isolate reward-related signals. However, most IC neurons exhibit motor responses during key presses (as indicated in Fig.6), which can complicate distinguishing between reward-related activity and motor responses. This complexity is why we initially presented neurons without motor responses. To clarify this point, we have added sentences in the Results section to explain the rationale behind our selection of neurons and to address the potential overlap between motor and reward responses in the IC.

"This phenomenon was further supported by examining the responses in the duration deviation detection task. Since most IC neurons exhibit motor responses during key presses (Supplementary Figure 6), which can complicate distinguishing between reward-related activity and motor responses, we specifically selected two neurons without motor responses during key presses (Figure 5)." (P.13, Line. 10-15)

(2) Addition of Data on Key Press Errors and No-Response Trials:

In response to the reviewer's suggestion, we have demonstrated Peri-Stimulus Time Histograms (PSTHs) for two example neurons during error trials as below, including incorrect key presses and no-response trials. Given that the monkeys performed the task with high accuracy, the number of error trials is relatively small, especially for the control condition (as shown in the top row of the figure below). While we remain cautious in drawing definitive conclusions from this limited trials, we observed that no clear reward signals were detected during the corresponding window (typically centered around 150 ms after the end of the sound). It is important to note that the experiment was initially designed to explore decision-making signals in the IC, rather than focusing specifically on reward processing. However, the data in Fig. 6 demonstrated intriguing signals of reward prediction error, which is why we believe it is important to present them.

When combined with the results from our reward experiment (Fig. 5), we believe these findings provide compelling evidence of reward prediction errors being processed by IC neurons.

**Author response image 1. sa3fig1:** (**A**) PSTH of the neuron from Figure 5A during a key press trial under control condition.The number in the parentheses in the legend represents the number of trials for control condition. (**B**) PSTHs of the neuron from Figure 5A during non-key press trials under experimental conditions. The numbers in the parentheses in the legend represent the number of trials for experimental conditions. (C-D) Equivalent PSTHs as in **A-B** but from the neuron in Figure 5B.

We are grateful for the reviewer's insightful suggestions, which have allowed us to improve the depth and rigor of our analysis. We believe these revisions significantly enhance our manuscript's conclusions regarding the complex functionalities of IC.

**Recommendations for the authors:**

**Reviewer #1 (Recommendations for the authors):**
One of the major issues of this work is that its writing fails to convey the focus and significance of the work. Sentences are too long and multiple pieces of information are often integrated in one sentence, causing great confusion.

We appreciate the reviewer's feedback regarding the clarity and structure of the manuscript. We agree that scientific writing should be clear and concise to effectively communicate the significance of the work. In response to this comment, we have undertaken the following revisions to improve the readability and focus of the manuscript:

(1) Simplified Sentence Structure:

We have revisited the manuscript and revised sentences that were overly complex or contained multiple pieces of information. Long sentences have been broken into shorter, more digestible statements to improve clarity and readability. Each sentence now conveys a single, focused idea.

(2) Improved Flow and Focus:

We have restructured certain paragraphs to ensure that the narrative flows logically and highlights the key findings. This restructuring includes placing the most significant results in prominent positions within paragraphs and ensuring that each section begins with a clear statement of purpose.

"Building upon the findings from the deviant responses, we next explored whether the climbing effect also manifested in responses to preceding standard stimuli, thereby examining the influence of sensory prediction and repetition on IC neuronal activity." (P.9, Line. 5-7)

"To determine whether the observed climbing effect was driven by reward anticipation, we designed an experiment controlling for reward effects, thereby clarifying the underlying factors influencing IC neuronal activity." (P.10, Line. 22; P.11, Line. 1-2)

"Recognizing that some IC neurons responded to reward delivery, we investigated whether these responses reflected reward prediction errors, thereby further elucidating the IC's role in reward processing." (P.12, Line. 9-11)

"Finally, to determine whether the IC plays a role in decision-making processes related to auditory perception, we analyzed the correlation between neuronal activity and behavioral choices in the duration deviation detection task." (P.14, Line. 4-6)

(3) Refined Significance of the Work:

In response to the reviewer's concern that the manuscript fails to clearly convey the significance of the work, we have revised the Introduction and Discussion sections to better emphasize the focus and impact of our findings. We now explicitly highlight the novel contributions of this research to the understanding of the multifaceted role of the IC in sensory prediction, decision-making, and reward processing.

"In this research, we embarked on a deviation detection task centered around sound duration with trained monkeys, performing extracellular recordings in the IC. Our observations unveiled a 'climbing effect'—a progressive increase in firing rate after sound onset, not attributable to reward but seemingly linked to sensory experience such as sensory prediction. Moreover, we identified signals of reward prediction error and decision-making. These findings propose that the IC's role in auditory processing extends into the realm of complex perceptual and cognitive tasks, challenging previous assumptions about its functionality." (P.6, Line. 1-8)

"Overall, our results strongly suggest that the inferior colliculus is actively engaged in sensory experience, reward prediction and decision making, shedding light on its intricate functions in these processes." (P.16, Line. 10-12)

We believe these revisions address the reviewer's concern and will make the manuscript more accessible to readers. Thank you for the valuable suggestion, which has led to a more precise and effective presentation of our work.

**Reviewer #2 (Recommendations for the authors):**
(1) In oddball paradigm, inter-stimuli-interval of 0.6 seconds was used. Vary the inter-stimulus-interval should prove whether this effect is rhyme learning. It is better to choose random inter-stimuli-interval and inter-trial-interval for each experiment across whole experiment in case monkeys try to remember the rhythm.

The reviewer suggests that the fixed inter-stimulus interval (ISI) of 0.6 seconds may lead to a rhythmic auditory response, allowing monkeys to anticipate sounds. This is a valuable suggestion, and we appreciate this perspective. However, we believe that rhythm is unlikely to play a significant role in driving the 'climbing effect.' The 'climbing effect' starts as early as the second sound in the block (as shown in Fig. 2D and Fig. 3B), which is before any rhythm or pattern could be fully established. Typically, rhythm learning requires at least three repetitions to form a predictable sequence.

Unfortunately, we did not vary the inter-stimuli-interval in the current study, so we cannot directly test this hypothesis with the current dataset. However, we agree with the reviewer that using random ISIs would be an effective way to rule out any potential contribution of rhythm learning to the climbing effect directly.

(2) Regarding "reward effect" on IC neurons' responses, we should rule out the possibility of simple auditory response to the switching of electromagnetic valve.

We appreciate the reviewer's concern about the potential confounding factor of the electromagnetic valve's click sound during water reward delivery, which could be interpreted as an auditory response rather than a true reward-related response. Anticipating this issue, we took measures to eliminate this possibility by placing the electromagnetic valve outside the soundproof room where neuronal recordings were conducted. This setup ensured that any potential auditory noise from the valve was minimized and unlikely to influence the IC neuronal activity.

To address this concern more explicitly, we have added a description in the Methods section detailing this setup. This revision clarifies the steps we took to rule out this potential confound, strengthening the validity of our claim that the observed IC activity is genuinely related to reward processing and not a simple auditory response to the valve's operation.

We thank the reviewer for bringing attention to this critical aspect of our experimental design, and we hope this clarification enhances the interpretation of our findings.

"The reward was controlled electronically by a valve located outside the sound-proof room to prevent any noise interference from the valve." (P.24, Line. 6-7)

(3) Since monkeys are smart, simple Go/NoGo design is not a good strategy. The task with more buttons to press, such as 2-AFC or 4-AFC task, may prevent artificial effect of unwanted behaviors and offer us more reliable and useful data.

We appreciate the reviewer's suggestion to implement a more complex behavioral task, such as a 2-Alternative Forced Choice (2-AFC) or 4-AFC design, to reduce the possibility of unwanted behaviors and to gather more reliable data. We agree that such paradigms could offer additional insights and help control the monkeys' decision-making processes by reducing potential confounding factors related to the simplicity of Go/NoGo responses.

In our current study, we chose the Go/NoGo task because it aligns with our primary experimental goal: investigating the relationship between IC activity and sensory prediction, decision-making, and reward processing in a simplified manner. This task allowed us to focus on reward prediction and sensory responses without introducing additional complexity that could increase the cognitive load on the monkeys and affect their performance. It is worth noting that training monkeys to perform auditory tasks is generally more challenging compared to visual tasks, though they are indeed capable of complex learning.

Moreover, this novelty detection task was initially designed as an oddball paradigm to explore predictive coding along the auditory pathway. Our lab has concentrated on this topic for several years, with the majority of current research focusing on non-behavioral subjects such as rodents. Implementing a more advanced paradigm like 2-AFC would have increased training time and required a different approach than our core objective.

That said, we agree that future studies would benefit from using more sophisticated tasks, such as 2-AFC or 4-AFC paradigms, as they could offer a more refined understanding of decision-making processes while enhancing the quality of data by minimizing unwanted behaviors. We believe that incorporating more advanced behavioral paradigms in future work will further enhance the rigor and reliability of our findings.

(4) Line 52, "challenges...", sounds a little bit too much. The authors tried to sell the ideal that IC is more than simple sensory relay point. I agree with that and I know the experiments on monkeys are not easy to gain too much comprehensive data. But to support authors' further bold opinions, more analysis is need to be done.

We appreciate the reviewer's feedback on the tone of the statement in Line 52, where we describe the findings as "challenging" conventional views of the IC as a simple sensory relay point. We agree that while our data provides intriguing insights into the multifunctionality of the IC, especially in sensory prediction, decision-making, and reward processing.

To address this, we have toned down the language in the revised manuscript to better reflect the current state of our findings. Rather than presenting the results as a direct challenge to existing knowledge, we now describe them as contributing to a growing body of evidence that suggests the IC plays a more integrative role in auditory processing and cognitive functions.

"This research highlights a more complex role for the IC than traditionally understood, showcasing its integral role in cognitive and sensory processing and emphasizing its importance in integrated brain functions." (Abstract, P.3, Line.12-15)

"This modulation by preceding sensory experiences indicates that the IC is more than merely a relay station, suggesting a more intricate role in auditory processing influenced by both ascending and descending neural pathways." (P.17, Line. 3-5)

(5) Line 143, "peak response", it is better not to refer this transient response as "peak response". How about "transient response" or "transient peak response"?

Thank you for your suggestion regarding the terminology used in Line 143. We agree with the reviewer that referring to this as simply a "peak response" could be misleading. To improve clarity and precision, we have revised the term to "transient peak response" as recommended.

We believe this adjustment better captures the nature of the neuronal activity observed and avoids confusion. The manuscript has been updated accordingly, and we appreciate the reviewer's valuable input.

(6) Is it possible to manipulate IC area and check the affection in behavior task?

We appreciate the reviewer's suggestion to manipulate the IC area and observe its effect on behavior during the task. Indeed, this would provide valuable causal evidence regarding the role of the IC in sensory prediction, decision-making, and reward processing, which would complement the correlational findings we have presented.

However, in this particular study, we focused on electrophysiological recordings to observe naturally occurring neuronal activity in behaving monkeys. While it is certainly feasible to manipulate IC activity, such as through pharmacological inactivation, optogenetics, or electrical stimulation, these techniques pose technical challenges in primates. Moreover, manipulating the IC, given its role as a lower-level relay station in the auditory pathway, could potentially disrupt auditory processing more broadly, complicating the interpretation of behavioral outcomes.

That said, we agree that introducing such manipulations in future studies would significantly enhance our understanding of the causal role of the IC in cognitive and sensory functions. We have now emphasized this as a key future research direction in the revised manuscript's discussion section. Thank you for this insightful suggestion.

"Further research is required to explore the underlying neuronal mechanisms and functional significance of this dynamic change comprehensively." (P.18, Line. 11-12)

**Reviewer #3 (Recommendations for the authors):**
Minor Comments:(1) Figure Labeling:The figures require more precise labeling, particularly concerning the analysis time windows, to facilitate reader understanding of the results.

We thank the reviewer for highlighting the importance of precise figure labeling, particularly regarding the analysis time windows. We understand that clear labeling is critical for conveying our findings effectively.

In response to your suggestion, we have revised the figures to include more precise and detailed labels, especially for the analysis time windows. These changes will help guide readers through the experimental design and clarify the interpretation of the results. We hope these improvements enhance the overall clarity and accessibility of the figures.

(2) Discrepancies in Figures and Text:There are discrepancies in the manuscript that could confuse readers. For example, on line 154, what was referred to as Supplementary Figure 1 seemed to actually be Supplementary Figure 2. Similar issues were noted on lines 480 and 606.

We appreciate the reviewer bringing this issue to our attention. We apologize for the discrepancies between the figures referenced in the text and their actual labels in the manuscript, as this could indeed confuse readers.

We have carefully reviewed the entire manuscript and corrected all discrepancies between the figures and their corresponding references in the text, including the issues noted on lines 154, 480, and 606. We have ensured that the figure and supplementary figure references are now consistent and accurate throughout the manuscript.

(3) Inconsistent Formatting in Figure legends:Ensuring a more professional and uniform presentation throughout the manuscript would be appreciated. There was inconsistent use of uppercase and lowercase letters in legends.

We appreciate the reviewer's attention to detail regarding the formatting of figure legends. Ensuring a professional and consistent presentation is crucial for enhancing the readability and overall quality of the manuscript.

We have carefully reviewed all figure legends and made the necessary corrections to ensure consistent use of uppercase and lowercase letters, as well as uniform formatting throughout the manuscript. This includes ensuring that all abbreviations and terminology are used consistently across the text and legends.